# Sustainable Management of Major Fungal Phytopathogens in Sorghum (*Sorghum bicolor* L.) for Food Security: A Comprehensive Review

**DOI:** 10.3390/jof11030207

**Published:** 2025-03-06

**Authors:** Maqsood Ahmed Khaskheli, Mir Muhammad Nizamani, Entaj Tarafder, Diptosh Das, Shaista Nosheen, Ghulam Muhae-Ud-Din, Raheel Ahmed Khaskheli, Ming-Jian Ren, Yong Wang, San-Wei Yang

**Affiliations:** 1Department of Plant Pathology, College of Agriculture, Guizhou University, Guiyang 550025, China; khaskheli.maqsood89@gmail.com (M.A.K.); mirmohammadnizamani@outlook.com (M.M.N.); entajmycology@gmail.com (E.T.); gm3085pp@outlook.com (G.M.-U.-D.); renmj72@163.com (M.-J.R.); 2Molecular and Applied Mycology and Plant Pathology Laboratory, Centre of Advanced Study, Department of Botany, University of Calcutta, 35, Ballygunge Circular Road, Kolkata 700019, West Bengal, India; diptoshmycology@gmail.com; 3Department of Food and Animal Sciences, Alabama A&M University, Normal, AL 35762, USA; shaista.nosheen@aamu.edu; 4Department of Plant Pathology, Faculty of Crop Protection, Sindh Agriculture University, Tandojam 70060, Pakistan; ra3715285@gmail.com

**Keywords:** fungal phytopathogen management, crop rotation, disease-resistant varieties, digital agriculture, predictive modeling

## Abstract

Sorghum (*Sorghum bicolor* L.) is a globally important energy and food crop that is becoming increasingly integral to food security and the environment. However, its production is significantly hampered by various fungal phytopathogens that affect its yield and quality. This review aimed to provide a comprehensive overview of the major fungal phytopathogens affecting sorghum, their impact, current management strategies, and potential future directions. The major diseases covered include anthracnose, grain mold complex, charcoal rot, downy mildew, and rust, with an emphasis on their pathogenesis, symptomatology, and overall economic, social, and environmental impacts. From the initial use of fungicides to the shift to biocontrol, crop rotation, intercropping, and modern tactics of breeding resistant cultivars against mentioned diseases are discussed. In addition, this review explores the future of disease management, with a particular focus on the role of technology, including digital agriculture, predictive modeling, remote sensing, and IoT devices, in early warning, detection, and disease management. It also provide key policy recommendations to support farmers and advance research on disease management, thus emphasizing the need for increased investment in research, strengthening extension services, facilitating access to necessary inputs, and implementing effective regulatory policies. The review concluded that although fungal phytopathogens pose significant challenges, a combined effort of technology, research, innovative disease management, and effective policies can significantly mitigate these issues, enhance the resilience of sorghum production to facilitate global food security issues.

## 1. Introduction

Sorghum (*Sorghum bicolor* L.), commonly known as great millet or durra, is a staple food crop that is particularly important in the semi-arid and arid regions of Africa, Asia, and Central and South America [1,2]. It ranks fifth globally among the most important cereal crops, followed by rice, wheat, maize, and barley. The USA is the leading producer of sorghum in the world, accounting for around 13% of the total output. Nigeria, Sudan, Mexico, and Ethiopia are the next leading producers, accounting for 11%, 8%, and 8%, respectively [3,4]. India plays a significant role in Asia, contributing a large share to global production. In Central America, Mexico is the leading producer, while South America’s main contributors include Argentina, Brazil, and Venezuela [5]. Sorghum is drought-resistant and can thrive under conditions unsuitable for several other crops, making it a vital food security crop in numerous regions prone to dry spells [6,7]. Economically, sorghum is essential for subsistence in developing countries and as a cash crop in both developing and developed countries. It has broad applications in human food, animal feed, and industry [8,9]. Sorghum grains are used in food products such as porridge, unleavened bread, cookies, cakes, couscous, and malted beverages; they are also used in the production of alcoholic beverages [10]. Moreover, sorghum serves as a feed grain for livestock and is increasingly used for ethanol production and bioenergy, owing to its favorable energy balance [11].

Globally, sorghum plays a significant role in agriculture, with diverse applications. Africa, the largest producer, contributes to approximately half of the global sorghum production, primarily as a food staple in the Sahel region, including Sudan, Nigeria, Ethiopia, Burkina Faso, and East African countries like Uganda and Tanzania [12,13,14]. In Asia, India leads the production, followed by China, where sorghum is used for food, fodder, and alcohol production, a trend also observed in Pakistan, Indonesia, and the Philippines [15,16]. In the Americas, the United States is a key player in the global sorghum market, using it primarily for animal feed and ethanol production, with major cultivation in the Great Plains and notable contributions from Mexico, Argentina, and Brazil [17,18]. In Australia, sorghum is the third-largest cereal crop, primarily grown in Queensland and northern New South Wales as livestock feed [19]. These regional differences highlight sorghum’s versatility and global importance in addressing agricultural and economic needs, while emphasizing the need to understand *Colletotrichum sublineolum* biology, improve disease forecasting, and develop sustainable disease management strategies [20,21].

The global sorghum market was valued at approximately USD 22.11 billion in 2023 and is anticipated to grow at a compound annual growth rate (CAGR) of 5.4% from 2024 to 2030, reflecting its economic significance [22]. Meanwhile, the global millet market was valued at USD 13.84 billion in 2023 and is projected to reach USD 21.20 billion by 2031, growing at a CAGR of 5.52% from 2024 to 2030 [23]. Market growth is primarily driven by the increasing consumption of sorghum in the feed and food industries, the rising demand for gluten-free food products, and the expansion of biofuel and ethanol production [24,25]. Because of its substantial economic importance, several factors that reduce sorghum yield, such as fungal diseases, pose serious concerns for farmers, industries, and policy makers [26].

Fungal diseases pose a significant challenge to sorghum production worldwide. They cause dramatic yield losses and also adversely affect the quality of the harvested grains. Yield losses owing to fungal infections can range from 10 to 30%, with some cases of complete crop failure under heavy disease pressure [27,28]. The nature and extent of damage caused by fungal diseases depend on several factors, including the specific fungal pathogen, susceptibility of the sorghum variety, environmental conditions, and timing and severity of disease onset [29]. Some of the most damaging fungal diseases in sorghum include anthracnose, grain mold, charcoal rot, downy mildew, and rust [30].

Anthracnose caused by *Colletotrichum* spp., especially *Colletotrichum sublineolum*, can cause severe yield losses. Grain mold, caused by a complex of fungi, including *Fusarium* spp., and blight, caused by *Curvularia* spp., primarily affect grain quality by reducing both its nutritional value and marketability [31,32]. Charcoal rot, caused by *Macrophomina phaseolina*, leads to stalk rot, which can significantly reduce the yield, especially under drought and high-temperature conditions [33]. The adverse effects of fungal diseases highlight the need for effective disease management. Managing sorghum fungal disease is complicated due to the diversity of pathogens involved, variability in the environmental conditions, and limited available resources, particularly in developing regions where sorghum is a primary food crop [12,34]. Despite advancements in breeding disease-resistant varieties and the availability of fungicides, these challenges persist, making fungal diseases a pressing problem in sorghum production [15].

Fungal phytopathogens that occur during sorghum production can have profound economic consequences and affect farmers, communities, and related industries. These diseases have a direct effect on crop yield, with significant losses leading to reduced income for farmers [35,36,37]. This reduction in yield is especially detrimental for smallholder farmers, who often depend on crops for their livelihoods. Consequently, poor harvests can have far-reaching financial implications for farmers and their families [38]. The quality of sorghum grain is a critical factor. Diseases such as grain molds can severely degrade grain quality, reducing the grain market value and suitability for specific end uses, which can restrict marketing opportunities and further diminish the potential earnings of farmers. The management of fungal phytopathogens also leads to increased production costs [39]. This may involve the purchase and application of fungicides, investment in resistant cultivars, or the implementation of labor-intensive cultural practices, such as crop rotation and sanitation. These additional costs can place significant financial strain on farmers, particularly those with limited resources [40,41]. Food security is crucial, particularly in regions where sorghum is the staple food source. Yield losses due to fungal diseases can lead to food insecurity, which is a grave concern in areas where alternative food sources are scarce or expensive [42]. The broader economic resilience of farming communities poses risks. Disease outbreaks can erode savings, increase debt, and heighten vulnerability to shocks like extreme weather or market fluctuations [43]. Moreover, the impact of sorghum disease extends beyond agricultural industries, such as milling, brewing, and animal feed production, which depend on sorghum and can suffer from reduced supply, potentially leading to job losses and economic downturns in these sectors [39].

This review provides a comprehensive analysis of major fungal diseases affecting sorghum, focusing on their impact, pathogenesis, symptomatology, and broader economic, social, and environmental implications. The diseases covered include anthracnose, grain mold complex, charcoal rot, downy mildew, and rust, with an emphasis on their pathogenesis, symptomatology, and overall economic, social, and environmental impacts. Current management strategies, such as fungicide application, biological control measures (BCMs), crop rotation, intercropping, and breeding for disease resistance, have been evaluated. Additionally, the review highlights emerging technologies like digital agriculture, predictive modeling, remote sensing, and IoT devices for early warning, detection, and disease management. It also outlines crucial policy recommendations to support farmers and research disease management, emphasizing the need for increased investment in research, strengthening extension services, facilitating access to necessary inputs, and implementing effective regulatory policies. Fungal diseases continue to pose significant challenges to sorghum production systems, threatening global food security and agricultural sustainability. Addressing these issues requires a comprehensive understanding of the pathogens involved, their biology, and the environmental factors driving disease outbreaks. This review provides an in-depth analysis of the current knowledge on fungal diseases affecting sorghum, highlighting recent advancements in disease management strategies, technological innovations, and policy interventions.

## 2. Overview of Sorghum Cultivation

### 2.1. Distribution of Sorghum Production Worldwide

Sorghum is cultivated worldwide, predominantly in warm, and semi-arid regions due to its drought tolerance and ability to grow in nutrient-poor soils where other cereal crops may not thrive [16,44]. Sorghum is a cornerstone of agricultural production in Africa, particularly in the Sahel region. Africa contributes to approximately half of global sorghum production, underscoring its significance as a primary staple food [45]. Countries such as Sudan, Nigeria, Ethiopia, and Burkina Faso are major producers that rely heavily on this crop for food security [28]. East African nations, such as Uganda and Tanzania, also cultivate significant amounts of sorghum [17]. Asia also plays a crucial role in global sorghum production, with India leading, followed by China [19]. Sorghum serves multiple purposes in these regions; it is a staple food, a source of fodder for livestock, and a key ingredient in alcohol production. Other Asian countries, such as Pakistan, Indonesia, and the Philippines, also contribute to sorghum production in the region [46]. The United States is the leading producer and one of the largest global sorghum exporters. The crop is primarily used for animal feed and ethanol production in the Great Plains region, including states such as Kansas, Texas, and Oklahoma, which are major production areas [47,48]. Sorghum is grown in significant quantities in Mexico, Argentina, and Brazil [49,50]. Sorghum is the third-largest cereal crop in Australia, after wheat and barley, and is primarily cultivated in Queensland and northern New South Wales [51]. It is primarily used as feed grain for livestock and plays a crucial role in the country’s agricultural sector. The approach of each region to sorghum cultivation reflects its agricultural practices, economic needs, and climatic conditions, making sorghum a globally significant crop with diverse applications [52]. All papers published on sorghum from 2000 to April 2024 have been analyzed across various topics. The distribution of research topics highlights significant areas of focus within sorghum studies. Crop rotation emerged as the most studied topic, with 851 publications, followed by environmental impacts, with 525 papers. Fungal disease management accounted for 63 publications, while digital agriculture management had the least, with only 12 publications. Other specialized areas, such as biological control (53 papers), fungicide development (22 papers), disease-resistant varieties (33 papers), and predictive modeling (39 papers), showed targeted research efforts. The distribution of publications over the years indicates a gradual increase in research interest, with notable spikes in some years. The analysis demonstrates a steady rise in publications on topics such as crop rotation and environmental impacts, whereas more recent topics like digital agriculture management have fewer studies, reflecting emerging areas of focus (Figure 1).

### 2.2. Primary Production Practices and Their Relation to Disease Prevalence

Sorghum production practices vary widely across regions, depending on local climatic conditions, soil types, available resources, and the intended use of the crop. These practices can significantly influence the prevalence and severity of fungal phytopathogens in sorghum. Understanding production practices and their impact on disease prevalence is crucial for developing effective disease management strategies. By adapting production practices, farmers can create fewer favorable conditions for fungal growth, reducing the severity and impact of diseases on sorghum yield (Table 1).

### 2.3. Importance of Fungal Phytopathogen Management in Sorghum Farming Systems

Fungal phytopathogen management is crucial to sorghum farming systems for several reasons. Table 2 provides a concise overview of the critical reasons for managing fungal phytopathogens in sorghum and highlights the key points, implications, and references for further information. This underscores the significance of this management in various aspects, such as yield protection, food security, environmental stewardship, and economic stability.

## 3. Main Fungal Diseases in Sorghum

### 3.1. Anthracnose in Sorghum

Anthracnose, caused by *Colletotrichum sublineolum*, is one of the most destructive foliar diseases of sorghum, particularly in tropical and subtropical regions, where warm temperatures and high humidity favor its development. This disease can cause yield losses of up to 67% in susceptible cultivars under favorable environmental conditions [30]. Anthracnose affects all above-ground parts of the plant and progresses through four distinct stages: root rot during the seedling stage, leaf and sheath lesions, stalk rot, and grain mold. These stages reflect the pathogen’s ability to infect and damage various parts of the sorghum plant, leading to significant agronomic and economic impacts [30].

At the seedling stage, anthracnose typically causes root rot, which weakens young plants and hinders early development. As the disease advances, the leaf and sheath phase is marked by characteristic dark, circular to elliptical spots on leaves. These lesions often coalesce into larger necrotic areas, reducing photosynthetic capacity and overall plant vigor. The pathogen can also invade the stalks, causing stalk rot that compromises structural integrity and may lead to peduncle breakage, resulting in grain loss. Infected grains exhibit symptoms ranging from discoloration to complete degradation by fungal growth [30,76,77,78].

The symptoms of anthracnose are visible as dark lesions on the leaves, often accompanied by red or orange pigmentation. These lesions may expand into necrotic patches that reduce photosynthesis. Infected stems may develop soft rot, weakening the plant’s structure and causing panicle collapse, which adversely affects grain development. In severe cases, anthracnose significantly reduces grain quality by causing premature grain deterioration and mold formation. Grain infected by *C. sublineolum* is often lightweight, with reduced nutritional value, impacting marketability and food security [79].

The pathogen follows a biotrophic-to-necrotrophic lifecycle. It establishes a symptomless biotrophic phase in host tissues before transitioning to a necrotrophic phase characterized by cell death and fungal proliferation [80]. Acervuli develop within lesions during this phase, producing masses of conidia that serve as secondary inoculums for further spread [77,81]. The pathogen’s genetic variability complicates management efforts, as different pathotypes can overcome resistance in sorghum varieties over time [78].

Additionally, certain growth stages of the plant are more vulnerable to specific pathogens, which has important implications for disease management. Anthracnose, caused by *Colletotrichum sublineolum*, is characterized by small oval to irregular reddish-brown to dark brown spots on the leaves and stems. These spots can enlarge over time, leading to premature leaf death. Lesions on stems and panicles may progress to rotting [82,83]. The disease can be identified by the presence of dark fungal structures (acervuli) within lesions, which may produce orange spores under wet conditions [84]. Anthracnose can infect sorghum at any stage, but symptoms are most commonly observed during the vegetative and early reproductive stages. Infections during these stages can result in significant yield losses [85].

### 3.2. Grain Mold Complex (Fusarium spp., Curvularia spp., and Others)

Grain mold complex is a significant sorghum disease, particularly in regions with high humidity and rainfall during the grain-filling stage [31]. It involves a group of diverse fungal pathogens, with the most common being *Fusarium*, *Curvularia*, and others such as *Aspergillus* and *Penicillium* [30,86]. Fungi involved in the grain mold complex have similar life cycles. They typically overwinter in soil or plant debris and produce spores that are dispersed by wind or rain. The spores germinate and infect the developing grain under favorable conditions, including high humidity and temperatures between 25 and 30 °C [87]. Grain mold is characterized by fungal growth on the grain surface, which can be of different colors (pink, green, black, or white) depending on the fungi involved. Infected grains can become discolored, shrimp, lightweight, or completely consumed by fungi. In severe cases, fungal growth can cover the entire panicle, resulting in a moldy appearance [88]. The symptoms of the grain mold complex are depicted in Figure 2, showing fungal hyphae enveloping grain particles. Grain molds primarily affect the quality of harvested grains, thereby reducing their market and nutritional value. Lightweight or shriveled grains are less desirable for milling, brewing, and other food or feed uses, leading to economic losses for farmers [89]. This disease is mainly associated with poorer grain filling, resulting in reduced grain weight and ultimately yield losses [90].

The grain mold complex, caused by fungi such as *Fusarium* spp., *Curvularia* spp., and others, primarily affects the grains. Symptoms include discoloration and shriveling of grains, which may appear white, pink, or black, often with a lightweight and chalky texture [91]. The presence of mold fungi is confirmed by spore masses in various colors, including white, green, pink, or black [92]. This disease predominantly affects the grain-filling stage, beginning at flowering and continuing until harvest. Moldy grains lead to significant yield losses and reduced quality [93]. Management strategies for grain mold complexes in sorghum include genetic resistance, cultural practices, and chemical control. Resistant varieties are a key part of disease management, although achieving complete resistance is challenging due to the complex nature of the disease and the availability of only partial resistance in current varieties [78]. Cultural practices that can help manage the disease include crop rotation, especially with non-host crops, and residue management to reduce the inoculum load in the field [79,80]. Timely harvesting can also help minimize grain mold development. Fungicides can be effective but are often not economically feasible for small-scale farmers and may have negative environmental impacts [81]. Continued research on the complex interactions between fungi, host plants, and environmental conditions is necessary to develop more effective and sustainable management strategies for grain mold complexes in sorghum [94].

### 3.3. Charcoal Rot (Macrophomina phaseolina)

Charcoal rot, which is caused by the soil-borne fungal pathogen *Macrophomina phaseolina*, is a common and destructive disease in sorghum, particularly in hot and dry environments. *Macrophomina phaseolina* is a necrotrophic fungus that can survive in soil and plant residues, such as microsclerotia (resting structures), for several years [95,96]. When soil moisture and temperature conditions become favorable, typically during high temperature and soil moisture stress, microsclerotia germinate and produce hyphae that penetrate the plant root system [97]. This fungus colonizes the vascular tissue of plants, obstructs water and nutrient transport, and eventually leads to plant death. The fungus produces new microsclerotia in dying plant tissues, which return to the soil once the plant dies and decomposes, thereby completing the disease cycle [98]. Charcoal rot symptoms typically appear during the flowering and grain-filling stages, particularly under drought conditions. The initial symptoms include wilting, yellowing, and premature leaf death [99]. Internally, the base of the stalk shows a silver-gray discoloration, and when split open, numerous tiny black microsclerotia give rise to charcoal dust; hence, the name “charcoal rot”. In the advanced stages, the disease leads to stalk lodging (breaking) and rot, causing the entire plant to die. The affected plants may produce small, poorly filled heads [100,101]. Charcoal rot symptoms on sorghum plants are illustrated in Figure 3, showing characteristic discoloration and fungal growth in the stem tissues. Charcoal rot can cause significant yield losses, particularly in years with high temperatures and droughts. Depending on the disease severity and environmental conditions, losses can be substantial [102]. In addition to yield reduction, charcoal rot can also reduce grain quality, thereby affecting economic returns for farmers [103]. Charcoal rot is most severe during the reproductive stage, particularly during grain filling, and is aggravated by high temperatures and drought stress [104]. Management of charcoal rot in sorghum involves an integrated approach. Crop rotation with non-host crops and proper irrigation and fertilization to reduce plant stress can be helpful [105]. The use of disease-resistant varieties is an important strategy, although resistance to charcoal rot is complex, and no completely resistant varieties are currently available [106]. Furthermore, deep plowing of infected crop residues and a reduction in the inoculum in the upper soil layers can be beneficial. Chemical control options are limited and typically not economically feasible [107]. Biocontrol agents, including certain beneficial soil microbes, are currently being studied for their potential to manage this disease [108,109,110]. Given the impact and challenges of managing charcoal rot, further research is needed to better understand the pathogen, develop effective resistant varieties, and devise sustainable and economically viable control methods [111].

### 3.4. Downy Mildew (Peronosclerospora sorghi)

Downy mildew, caused by *Peronosclerospora sorghi*, can lead to significant yield losses in sorghum, particularly in tropical and subtropical regions. While losses can range from moderate (10–30%) to severe (50% or more), depending on various factors, implementing effective management strategies can substantially reduce the adverse impacts on sorghum production [112,113,114]. *Peronosclerospora sorghi* can survive in soil and plant debris, such as oospores, for several years. Oospores germinate under favorable conditions (high humidity and cool temperatures), producing sporangia that release motile zoospores [115]. These zoospores infect plants primarily through their roots and move systemically throughout the plant [116]. The pathogen colonizes plant tissue and eventually produces sporangia on the lower leaf surface, which release new zoospores that can spread to other plants, thereby completing the life cycle of the pathogen [117,118]. The most characteristic symptom of downy mildew in sorghum is the presence of white to pale purple downy growth, which consists of sporangia, on the lower surface of the leaves [113]. Symptoms include white to pale purple downy growth on the lower surface of leaves, yellowing, or death of leaf tissue, and stunted plants [119]. In severe cases, plants may fail to produce fertile heads. Downy growth composed of sporangia is a characteristic sign of the disease [120]. Symptoms may persist throughout the plant’s life, underscoring the importance of early detection [98]. Other symptoms include chlorosis (yellowing) or necrosis (death) of the leaf tissue, plant stunting, and a bushy or rosette-like appearance due to shortened internodes. In severe cases, infected plants may or may not produce sterile heads, leading to significant yield loss [121]. Downy mildew symptoms are shown in Figure 4, characterized by yellowing and fuzzy grayish patches caused by *Peronosclerospora sorghi*. Effectively reducing Downy mildew in sorghum requires a comprehensive, integrated management approach that includes resistant varieties, appropriate fungicide use, optimal cultural practices, biological controls, and vigilant monitoring. Addressing these aspects not only minimizes yield losses but also preserves the quality of harvested grains, ensuring economic viability for farmers and maintaining the nutritional and market value of sorghum products [76,122,123,124,125]. The impact of the disease can be particularly severe in regions with high humidity and cool temperatures, which are favorable for disease development [31,126]. The management of downy mildew on sorghum involves several strategies. Planting disease-resistant varieties, such as BTx623 (*BTx623*) and Hybrid SPH 1705, is a key part of the control efforts. Crop rotation, particularly with non-host crops, such as legumes, cereals, root crops, brassicas, and cover crops, and residue management to reduce the inoculum load in the field can also help manage this disease. [97,127,128]. Utilizing fungicides with effective active ingredients and diverse chemical groups for seed treatment is a vital component in managing downy mildew (*Peronosclerospora sorghi*) in sorghum. By selecting fungicides such as dithiocarbamates (e.g., thiram), phenylamides (e.g., metalaxyl), triazoles (e.g., propiconazole), strobilurins (e.g., azoxystrobin), carboxamides (e.g., fludioxonil), chlorothalonil, and phosphorous acid (e.g., fosetyl-Al), farmers can provide comprehensive protection against initial disease infections [129,130,131,132,133]. Further research is required to develop more effective and sustainable management strategies, including breeding more resistant varieties and developing biological control methods. Understanding the biology and epidemiology of *Peronosclerospora sorghi* is critical for predicting disease outbreaks and improving disease management practices [134,135].

### 3.5. Rust (Puccinia purpurea)

Rust, caused by the fungal pathogen *Puccinia purpurea*, is a common sorghum disease, particularly in warm and humid regions [136]. *Puccinia purpurea* is an obligatory biotrophic pathogen that requires a living host to complete its life cycle. This fungus produces urediniospores that are wind-dispersed and can initiate infection under favorable conditions [137,138]. Rust is characterized by numerous small, round to elongated raised pustules on the upper surface of leaves, leaf sheaths, and occasionally panicles and stalks. These pustules are initially reddish-brown, but turn black as they mature [139]. The disease is confirmed by rust-colored pustules filled with spores [140]. Rust can infect sorghum at all stages, but it is most severe during the early growth stages. Late-planted crops or continuous cropping systems are particularly vulnerable due to pathogen build-up [141]. During infection, the pathogen colonizes the host tissue and produces urediniospores, leading to a secondary infection cycle [142,143]. Rust disease in sorghum is easily identified by the presence of numerous small, round, and elongated pustules on the upper surface of leaves, leaf sheaths, and occasionally panicles and stalks. These pustules are initially reddish-brown (hence the name “rust”) but turn black as they mature [144,145]. In severe cases, heavy rust infestation can lead to leaf blight and early senescence, thereby reducing plant photosynthetic capacity. Rust disease symptoms are highlighted in Figure 5, displaying orange pustules indicative of fungal infection. Sorghum rust can cause significant yield losses, especially in susceptible varieties and under favorable environmental conditions. This disease reduces photosynthetic area, affecting plant vigor, grain filling, and, ultimately, yield. Under heavy disease pressure, downy mildew can drastically impair sorghum grain quality, affecting multiple parameters, from kernel weight and protein content to germination rates and marketability. Quantitative assessments indicate that yield reductions can reach up to 70%, while quality parameters such as protein content and germination rates may decline by 25% or more [82,146,147,148,149]. *Puccinia sorghi* is a fungal pathogen responsible for rust disease in sorghum, causing significant crop losses. The pathogen infects the leaves and stems of sorghum plants, with disease development heavily influenced by environmental conditions, particularly humidity and temperature, with the ideal range for infection being 15–25 °C [126,142,150]. Its distribution is global, especially prevalent in regions like the Corn Belt and East Africa, where it coexists with other sorghum diseases. The impact of *Puccinia sorghi* on yields can vary, but it generally leads to moderate losses in susceptible cultivars, particularly when combined with other fungal infections like anthracnose and mildew [151]. Effective management strategies include developing genetically resistant sorghum varieties and improved diagnostic techniques for early disease detection, emphasizing the importance of integrated disease management to mitigate its effects on global sorghum production [152]. Rust management in sorghum primarily relies on the use of resistant varieties and crop rotations. Several rust-resistant sorghum hybrids are available, and their use is the most effective and economical method for disease control [153]. Crop rotation with non-host crops can reduce the amount of inoculum required in the field. Timely planting can also help to avoid periods of high disease pressure. Chemical control with fungicides can be effective, but is typically used as a last resort owing to the costs and potential environmental impacts [154,155]. Because of the importance of sorghum and the potential severity of rust, ongoing surveillance and research are essential to monitor the evolution of the pathogen and to develop new resistant varieties. Predictive models based on weather patterns can also help forecast disease outbreaks and guide management decisions [29,156].

### 3.6. Other Significant Diseases: Brief Descriptions and Impacts

Although less prevalent, several other fungal phytopathogens have a significant effect on sorghum production. These fungal phytopathogens may be as significant as the primary diseases discussed and can substantially affect sorghum production. The management of these fungal phytopathogens typically involves similar strategies, including the use of resistant varieties, crop rotation, and agronomic practices to reduce disease pressure and plant stress. A brief description of these fungal phytopathogens is provided in Table 3.

### 3.7. Pathogenesis and Symptomatology

Each fungal pathogen interacts with sorghum plants in unique ways, leading to different disease symptoms and effects on the crop [85]. Table 4 provides a comprehensive overview of additional fungal diseases affecting sorghum, highlighting their key symptoms, potential impact on crops, and references for further information. Understanding how fungi affect plants is crucial for disease diagnosis and management strategies. Research on host–pathogen interactions, disease resistance, and environmental effects on disease development can help improve disease control strategies and reduce the impact of these diseases on sorghum production.

## 4. Mycotoxins

Sorghum is highly susceptible to contamination by mycotoxins, secondary metabolites produced by fungi such as *Aspergillus*, *Fusarium*, and *Alternaria* species. These toxic compounds pose serious health risks to humans and animals and adversely affect crop quality, yield, and economic value. The primary mycotoxins associated with sorghum are aflatoxins, fumonisins, zearalenone, and deoxynivalenol (DON), among others. Sorghum is particularly vulnerable to aflatoxin contamination in warm and humid conditions during post-harvest storage or periods of drought stress during cultivation. Aflatoxin B1 is the most toxic and frequently detected in sorghum grains, often exceeding permissible limits set by food safety authorities [94,163,164]. Fumonisins, especially fumonisin B1, are hepatotoxic and nephrotoxic and are implicated in esophageal cancer in humans. Environmental conditions such as high humidity and fluctuating temperatures significantly influence fumonisin contamination in sorghum [165,166]. While zearalenone is known for its estrogenic effects, DON, also called vomitoxin, causes severe gastrointestinal disorders in humans and animals. Studies have highlighted sporadic yet notable contamination of sorghum with these mycotoxins, particularly in temperate regions [150,167,168]. Mycotoxin contamination in sorghum has far-reaching implications for global food security and public health. Chronic exposure to these toxins is associated with liver cancer, immune suppression, and stunted growth in children. In livestock, mycotoxins reduce feed efficiency, impair reproduction, and increase susceptibility to diseases [169,170,171].

## 5. Current Strategies for Managing Fungal Phytopathogens

### 5.1. Use of Fungicides: Advantages and Drawbacks

Fungicides play an important role in the management of fungal diseases in sorghum and other crops [172]. However, their use encompasses both advantages and drawbacks. Fungicides play a crucial role in modern agriculture by offering several advantages in disease management. Their efficacy in significantly reducing disease severity has been well documented, with studies highlighting their role in protecting crop yield and ensuring grain quality, both critical for the economic viability of agriculture [89]. For sorghum, fungicides such as tebuconazole, propiconazole, and azoxystrobin have demonstrated high efficacy in controlling fungal diseases like anthracnose caused by *Colletotrichum sublineola* and the grain mold complex. The availability and accessibility of fungicides also present notable benefits. A diverse range of fungicides is commercially available, allowing the targeted management of specific diseases. In many regions, these fungicides are readily accessible to farmers, ensuring that effective disease control measures are within reach. Moreover, the flexibility of fungicides in terms of application methods, such as foliar sprays, seed treatments, or soil drenches, allows farmers to tailor disease management strategies to prevailing conditions. Seed treatment with metalaxyl is often used to manage seed-borne downy mildew pathogens like *Peronosclerospora sorghi* [173,174]. Another advantage of fungicides is their dual functionality, i.e., preventive fungicides stop fungal infection before it occurs, while curative fungicides can halt disease progression even after infection [175,176]. For instance, strobilurins, like azoxystrobin, exhibit both preventive and curative properties, enhancing their utility in sorghum disease management. This versatility makes fungicides an integral component of integrated pest management (IPM) strategies [177].

The development of new fungicides is an ongoing area of research driven by the need for more effective, less toxic, and environmentally friendly solutions to manage fungal diseases in crops such as sorghum [29]. Recent advancements in fungicide formulations have focused on developing products specifically tailored for managing fungal diseases in sorghum. Notably, combinations such as azoxystrobin with difenoconazole and azoxystrobin with epoxiconazole have been registered for controlling leaf blight in sorghum [178,179]. These formulations leverage the synergistic effects of combining different active ingredients, providing broad-spectrum efficacy against multiple pathogens. Azoxystrobin, a strobilurin fungicide, inhibits mitochondrial respiration in fungi, effectively stopping spore germination and mycelial growth [180]. Difenoconazole and epoxiconazole are triazole fungicides that inhibit ergosterol biosynthesis, which is essential for fungal cell membrane integrity. This dual-action approach not only enhances effectiveness but also helps in managing resistance by targeting different biochemical pathways [181].

Effectiveness against Pathogens: These new formulations have demonstrated high efficacy against key sorghum pathogens such as *Colletotrichum sublineolum* (anthracnose) and *Exserohilum turcicum* (leaf blight), significantly reducing disease incidence and severity [182,183]. Comparative trials have shown these combinations to outperform older fungicides, offering improved control under varying environmental conditions. Despite these advancements, it is important to consider the potential challenges. The development of new fungicides is a lengthy and expensive process with strict regulatory requirements to ensure safety and efficacy. There may also be issues related to costs, accessibility, and acceptability among farmers. Furthermore, similar to any fungicide, these new formulations should be used as part of an integrated pest management strategy to maximize their benefits and minimize their potential drawbacks, such as the development of resistance. Notable advancements are listed in Table 5.

The widespread use of fungicides in agriculture, which are beneficial for controlling fungal diseases, has led to several significant issues. The first is the development of resistance [175,185]. Frequent and over-reliant use of fungicides can result in the evolution of fungal populations that are resistant to these chemicals, thereby reducing their efficacy over time. This phenomenon necessitates the continuous development of new fungicides, which increases the cost and complexity of managing crop diseases [191]. Fungal phytopathogens and their management have various environmental impacts. Here, we summarize the potential effects of the management of fungal phytopathogens in agriculture, particularly through the use of fungicides, and present several environmental challenges and implications [175,192]. Fungicide use can lead to environmental contamination, potentially polluting water sources and harming non-target organisms, including beneficial soil microbes, insects, birds, and mammals [190]. Furthermore, the overuse of these chemicals can result in the development of resistant fungal strains, exacerbating this problem. Therefore, its impact on soil health is a critical issue. In addition, numerous fungi responsible for crop diseases can survive in soil or plant debris, creating a reservoir of diseases for future crops [185]. This affects soil health and limits the variety of crops that can be grown in the same field in subsequent seasons without increasing disease risk [193]. Biodiversity loss is a significant consequence in areas where crops such as sorghum are extensively cultivated. The disease pressures can lead to shifts toward more disease-resistant crops or different crops entirely, potentially reducing agricultural biodiversity. This loss of biodiversity can negatively affect ecosystem stability and resilience [144]. Moreover, the carbon footprint associated with fungal diseases is a growing concern in agriculture. The reduction in yield per hectare necessitates an increased land area for production, which can lead to deforestation or the conversion of natural habitats into agricultural land [194]. In addition, the production and application of fungicides contribute to CO_2_ emissions. Waste production is another issue in which infected crops that cannot be harvested or sold lead to waste. If this waste is not properly managed, it can exacerbate the spread of the disease [195]. Ecosystem services are also affected by reduced yields and crop losses. Sorghum is often used as a cover crop to prevent soil erosion, promote nutrient cycling, and suppress weeds [196]. The loss of these services due to disease can lead to long-term environmental degradation. Given these diverse and significant impacts, the importance of effective and sustainable disease management strategies has become clear [197]. This may involve the use of crop rotation-integrated pest management organic farming practices, or other methods that balance disease control with environmental sustainability [198].

One example is the development of resistance to triazoles, such as tebuconazole, in *Fusarium* populations. In a study conducted by Little et al. [182], isolates of *Fusarium* from sorghum fields were shown to exhibit reduced sensitivity to tebuconazole, resulting in higher disease severity even after fungicide application. Similar resistance issues have been observed in *Colletotrichum* spp., with certain strains becoming less responsive to strobilurin fungicides like azoxystrobin.

The effect of silicon (Si) on sorghum resistance to anthracnose in resistant (BR005) and susceptible (BR009) lines. Si increased in both lines, but only affected disease severity in the susceptible line, where higher Si levels correlated with reduced disease. Si had little impact on the resistant line [199]. The impact of Si and fungicide on sorghum anthracnose. Calcium silicate (CS) reduced disease severity by 39–42%, while fungicide further decreased it. Si levels in leaves increased with CS, and yields improved by 0.6 ton/ha with CS and 0.48 ton/ha with fungicide. The residual effect of CS in the soil also enhanced Si content and reduced anthracnose in the following season [200]. Si-treated plants showed higher Si deposition at infection sites, smaller acervuli, and increased activity of defense enzymes (peroxidases and polyphenol oxidases), along with higher anthocyanin concentrations. These results suggest that Si not only helps form a physical barrier against *Colletotrichum sublineolum* but also contributes to biochemical defense mechanisms, improving sorghum’s resistance to anthracnose [201].

The development of new fungicide formulations also considers environmental and health impacts. Modern formulations aim to minimize toxicity and environmental persistence compared to older products. For instance, systemic fungicides like azoxystrobin are designed to be absorbed by plant tissues, reducing runoff into aquatic systems and minimizing non-target organism exposure [202]. Studies indicate that newer fungicide formulations have a lower impact on non-target species, including beneficial insects and soil microorganisms. This is achieved through targeted application methods and improved formulation chemistry that enhances plant uptake while reducing environmental dispersion [203]. The strategic use of multi-site action fungicides like those combining strobilurins with triazoles helps mitigate resistance development. By employing diverse modes of action, these formulations reduce the selective pressure on fungal populations, thereby prolonging the efficacy of available fungicides [204,205].

The development of resistance is primarily driven by the frequent and excessive use of fungicides, which place selective pressure on fungal populations. This phenomenon underscores the importance of integrating fungicide use with other disease management practices to slow resistance development [175]. Moreover, studies have indicated that combining fungicides with biological control agents or cultural practices like crop rotation can mitigate the emergence of resistant strains. For instance, Pothiraj et al. [206] demonstrated that rotating fungicides with different modes of action and integrating resistant sorghum varieties significantly reduced the prevalence of resistant *Fusarium* strains.

Moreover, the environmental impact of fungicides is not well understood. These chemicals contribute to environmental pollution, affecting not only the targeted fungal pathogens but also non-target organisms. This can lead to decreased biodiversity and disruption of the ecological balance, raising concerns regarding the long-term sustainability of intensive fungicide use [207]. Fungicides such as azoxystrobin and tebuconazole can cause unintended harm to non-target organisms, including beneficial insects, soil microorganisms, and aquatic ecosystems. The long-term environmental consequences of fungicide residues are not fully understood, but studies have indicated that their persistence in soil and water can disrupt local biodiversity. Rizvi et al. [208], highlighted the adverse effects of repeated fungicide applications on soil microbial communities in sorghum fields, leading to altered nutrient cycling and reduced soil health.

Another dimension of the problem involves human health and economic aspects. Health risks are associated with fungicides, especially when they are not used, according to safety guidelines [209]. Exposure to fungicides like captan and maneb can pose health hazards to farmers during application. [210]. The financial burden of purchasing and applying fungicides is particularly pronounced for smallholder farmers. The cost of fungicides such as azoxystrobin or tebuconazole can be prohibitive, limiting access to effective disease control measures [211]. Kumar et al. [212], noted that residues of tebuconazole in sorghum grains exceeded the safe limits set by food safety authorities, posing potential health risks to consumers. This economic barrier may result in inadequate protection against diseases, thereby reducing overall crop productivity. Therefore, fungicides should be part of an integrated disease management strategy. Reliance on fungicides alone is not a sustainable solution; incorporating practices such as crop rotation, the use of resistant varieties like sorghum bicolor hybrid CSH 16, and cultural practices like field sanitation can significantly reduce disease pressure. Combining these methods with judicious fungicide application can enhance disease control while mitigating the risks of resistance, environmental pollution, and health hazards [172].

### 5.2. Biological Control Measures: Efficacy and Limitations

Biological control measures involve the use of living organisms or their by-products to suppress disease-causing agents, including fungi [173]. In the context of sorghum, specific biological control agents include bacterial species such as *Bacillus subtilis* and *Pseudomonas fluorescens*, fungal antagonists like *Trichoderma harzianum* and *Trichoderma viride*, and mycoparasitic fungi such as *Gliocladium virens* [174]. The efficacy of biological control agents in agriculture has several advantages, particularly in terms of disease management and environmental sustainability [176]. These agents can effectively reduce pathogen populations by competing for nutrients, occupying similar ecological niches, or producing pathogen-inhibitory substances, thereby offering a direct and effective means of disease control [177].

*Trichoderma* is one of the most widely studied genera for biological control in agriculture, including sorghum. Species such as *Trichoderma harzianum* and *Trichoderma viride* are known for their ability to suppress soilborne fungal pathogens like *Fusarium* spp. (causing root rot and seedling blight) and *Sporisorium sorghi* (causing smut). *Trichoderma* acts through mechanisms such as *mycoparasitism* (where it directly attacks fungal pathogens), competition for space and nutrients, and the secretion of secondary metabolites like gliotoxins and chitinases, which inhibit the growth of other fungi [213,214,215]. *Trichoderma* formulations are typically applied as soil drenches or seed treatments at concentrations ranging from 10^8^ to 10^9^ CFU (colony-forming units) per gram of soil or seed [216,217,218].

*Bacillus*-based BCAs, particularly *Bacillus subtilis*, *Bacillus amyloliquefaciens*, and *Bacillus thuringiensis*, have shown significant efficacy in controlling foliar and soilborne fungal pathogens in sorghum [219,220]. These bacteria produce antimicrobial peptides, such as bacillomycin and surfactin, which inhibit the growth of pathogens like *Colletotrichum* sub-lineolum (causing anthracnose) and *Alternaria* spp. (causing leaf spot). *Bacillus*-based agents are often applied as foliar sprays or incorporated into soil at concentrations of 10^6^ to 10^8^ CFU per mL or g, respectively [220,221].

*Pseudomonas* species, such as *Pseudomonas fluorescens*, are well known for their broad-spectrum antifungal activity, particularly against soilborne fungi like *Fusarium* and *Sporisorium*. *Pseudomonas* spp. produce a range of bioactive compounds, including hydrogen cyanide, phenazines, and proteases, which disrupt fungal growth. They are typically applied as soil drenches or foliar sprays at concentrations similar to *Bacillus* spp., with doses ranging from 10^7^ to 10^8^ CFU per mL [222,223].

### 5.3. Efficacy of Biological Control in Sorghum

The effectiveness of biological control agents in sorghum is influenced by several factors, including environmental conditions, the nature of fungal pathogens, and the method of application [224,225].

Efficacy against *Fusarium* spp.: *Trichoderma harzianum* has demonstrated significant efficacy in controlling *Fusarium* spp., which causes *Fusarium* root rot and seedling blight in sorghum. Studies by Miljaković et al. [226], Showed that the application of *T. harzianum* as a seed treatment at a concentration of 10^8^ CFU/g effectively reduced *Fusarium*-induced root rot by up to 70%. Similarly, *Bacillus subtilis* has been shown to reduce *Fusarium*-induced blight in sorghum by promoting plant growth and enhancing root development [57].

Efficacy against *Sporisorium sorghi* (Smut): Biological control of smut in sorghum caused by *Sporisorium sorghi* has been challenging, but *Trichoderma* spp. have shown promising results in reducing disease severity. In field trials, *T. viride* was applied as a soil drench at a rate of 2 kg/ha, significantly reducing smut incidence by up to 50% compared to untreated controls [227,228].

Efficacy against *Colletotrichum* spp. (Anthracnose): *Bacillus subtilis* and *Pseudomonas fluorescens* have demonstrated efficacy in controlling anthracnose in sorghum. *P. fluorescens* applied as a foliar spray at a concentration of 10^7^ CFU/mL was able to reduce anthracnose severity by up to 60%, according to studies by Wei et al. [57]. Regarding environmental impacts, biological control methods are generally less harmful than chemical controls, significantly reducing chemical residues in the environment and crops, which benefits both ecosystem health and food safety [191]. In addition, biological control can be used to manage fungicide resistance and provide an essential alternative to disease management strategies [182]. Finally, the sustainability of these agents, once established, can provide long-term disease control, thereby contributing to the sustainability of agricultural practices. Collectively, these factors demonstrate the efficacy of biological control methods in promoting healthier crops and a more sustainable agricultural environment [206].

Although they offer sustainable alternatives to chemical pesticides, biological control methods in agriculture have several limitations that affect their effectiveness and adoption [229]. The effectiveness of biological control agents can vary significantly, as they are influenced by environmental conditions, specific pathogens, and the crops involved, a concern highlighted by Segoli et al. [230]. Furthermore, these methods are often slower than their chemical counterparts, making them less effective against rapidly progressing or established diseases [231]. Establishing biological control agents in the field also presents challenges, particularly under unfavorable conditions, and some agents may be costly or difficult to produce and apply [232]. Regulatory challenges, particularly for genetically modified organisms, further complicate the deployment of certain biological agents. Additionally, a lack of awareness and technical expertise among farmers limits the adoption of these strategies [233]. These challenges highlight the need for ongoing research and development, and farmer education programs and extension services are critical to overcoming these barriers and promoting the effective use of biological control measures. Biological control (BC) measures are important components of integrated disease management programs [61]. However, for maximum effectiveness, it must be combined with other strategies, such as cultural practices, resistant cultivars, and chemical controls when necessary. Further research is needed to improve the consistency and scalability of biological control methods. Advances in formulation technologies, such as microencapsulation and biochar-based carriers, and the identification of novel agents with broader environmental adaptability, could enhance the reliability and adoption of biological control strategies in the sorghum production systems [234].

### 5.4. Crop Rotation and Intercropping: Benefits and Challenges

Crop rotation and intercropping are two important agricultural practices that help manage fungal diseases in sorghum [235]. Crop rotation involves growing different crops in sequential seasons to manage soil health and reduce pest and disease pressure. For sorghum, suitable rotational crops include legumes like soybeans or cowpeas, which fix nitrogen, and cereal crops such as maize or wheat. Intercropping sorghum with crops like peanuts, beans, or sunflowers can help suppress weeds and improve pest control, while cover crops like clover enhance soil fertility [236,237,238,239]. Crop rotation, a fundamental practice in sustainable agriculture, provides multiple benefits that are crucial for maintaining healthy crops and ecosystems [54]. It effectively disrupts the disease cycle by replacing a susceptible crop with a non-host crop, thereby breaking the life cycle of specific pathogens and reducing their populations in the field [126]. In addition, crop rotation significantly enhances soil health, with different crops contributing uniquely, such as adding organic matter and leguminous crops and enriching nitrogen levels through nitrogen-fixing bacteria. This leads to improved soil fertility and structure [240]. Crop rotation plays a pivotal role in pest management by controlling the populations of insects and nematodes, which contributes to the overall health and yield of crops. These collective benefits underscore the importance of crop rotation in promoting ecological balance, reducing reliance on chemical inputs and ensuring sustainable agricultural productivity [241].

Crop rotation is a key sustainable agricultural practice that faces several challenges that hinder its effective implementation. One primary concern is crop selection, which can be particularly challenging in regions with specific climatic or market constraints, necessitating careful consideration of the suitability of each crop [242]. In addition, crop rotation requires meticulous planning and management to ensure that each crop contributes positively to the health and productivity of the soil for subsequent crops [243]. Economic factors also play a crucial role in market demand, and price considerations for rotational crops can significantly influence the decisions of farmers to adopt this practice. These challenges underscore the need for strategic planning that balances environmental, agronomic, and economic considerations to optimize the benefits of crop rotation [244].

Intercropping is the simultaneous growth of two or more crops in the same field. Intercropping offers multiple benefits for sustainable agriculture [245]. Disease suppression is a notable advantage because certain crops in an intercropped system can naturally reduce fungal diseases by altering the microclimate, competing for resources, or producing pathogen-inhibiting substances [58,246]. In addition, the resource use efficiency of the intercropping systems improved. Different crops utilizing various resources, such as sunlight, water, and nutrients, can lead to more efficient overall usage and potentially higher yields [247]. Furthermore, intercropping aids in diversifying pest populations, thereby diminishing the impact of a single pest or disease [248]. This ecological approach to pest and disease management combined with optimized resource utilization underscores intercropping as a powerful strategy for enhancing agricultural productivity and sustainability [249,250].

Intercropping, which is the practice of growing multiple crops in close proximity, poses several challenges. Crop compatibility is a primary concern because certain crop combinations can lead to competition instead of cooperation, potentially diminishing the benefits of intercropping [251]. The complexity of the management of these systems compounds this issue. The differing water, fertilization, and pest management requirements of each crop add complexity to traditional monoculture farming [249]. Furthermore, the harvesting process in intercropping systems (IS) is complex. The labor-intensive nature of harvesting multiple crops, especially when they mature at different times, requires more labor and careful timing [65]. These challenges highlight the need for strategic planning and informed crop selection to implement intercropping effectively. Both crop rotation and intercropping are important components of integrated sorghum disease management strategies. However, they must be adapted to local conditions and carefully managed [41,252].

### 5.5. Breeding for Disease Resistance: Progress and Potential

Breeding for disease resistance involves the development of sorghum varieties that are resistant or less susceptible to specific fungal diseases [141]. This can be a highly effective and sustainable approach to disease management as it reduces the need for chemical fungicides and can help improve yield and grain quality [97,253]. Recent advancements in sorghum breeding have led to significant progress in the enhancement of disease resistance. Techniques such as marker-assisted and genomic selection have enabled breeders to efficiently identify and select genes or quantitative trait loci linked to disease resistance, thus refining the breeding process [254]. The practical impact of these advancements is evident in the development of new sorghum varieties with resistance to diseases such as anthracnose and downy mildew [255]. Concurrently, there has been a notable increase in our molecular understanding of the interactions between sorghum and its fungal pathogens, which is critical for identifying key resistance genes [157]. This combination of advanced breeding techniques and enhanced molecular knowledge marks a pivotal era in the development of disease-resistant sorghum, paving the way for more resilient agricultural practices [85].

The potential to enhance disease resistance in sorghum is driven by cutting-edge technologies and genetic resources. Genome editing, particularly using CRISPR/Cas9, offers a revolutionary approach for precise genetic modifications, enabling the targeted enhancement of disease resistance [42,256]. Complementing this, the significant genetic diversity within sorghum, including various landraces and wild relatives, provides a rich source for breeding disease-resistant varieties [257]. In addition, advancements in computational biology and machine learning have ushered in an era of predictive breeding and accelerated the development of resistant strains by analyzing vast datasets to predict the most effective breeding strategies [258]. Collectively, these approaches represent a multifaceted and promising pathway toward developing robust, disease-resistant sorghum crops, aligning technological innovation with the rich genetic heritage of the plant [259].

The development of disease-resistant crops is a challenge in agricultural science. Resistance durability is a primary concern, as pathogens can evolve to overcome genetic resistance, particularly if they rely on a single gene [260]. This is compounded by the complexity of breeding for resistance, especially for diseases controlled by multiple genes or quantitative trait loci, which is intricate and demanding [261]. In addition, the adoption of these resistant varieties by farmers is hindered by factors such as seed availability, cost, and preference for specific crop traits [262]. These challenges highlight the need for a multifaceted approach to agricultural research and practice that encompasses the biological aspects of resistance breeding and addresses the socioeconomic factors influencing crop adoption [263]. Disease resistance breeding is an important component of integrated sorghum disease management. Continued investment in research and development, extension services to promote the adoption of resistant varieties, and policies to support seed systems are crucial for realizing the full potential of this approach [188,264].

Genetic and genomic studies have played pivotal roles in breeding disease-resistant sorghum. They provide new insights into the genetic architecture of resistance traits and aid in the precise identification and manipulation of these traits in Table 6. Genetic tools are revolutionizing crop breeding for disease resistance. They provide new opportunities to increase the efficiency and precision of breeding programs, helping to develop new sorghum varieties that are more resilient to fungal diseases. However, it is also important to consider that the effective use of these tools requires significant resources and expertise and that developing and adopting genetically modified or genome-edited crops may face regulatory and acceptance challenges in some regions. Recent studies have identified several key genetic loci associated with anthracnose resistance in sorghum. For instance, a genome-wide association study (GWAS) identified 38 loci significantly associated with anthracnose resistance, including genes encoding receptor-like kinases and nucleotide-binding leucine-rich repeats (NLRs), which are crucial for disease resistance [265]. Another study characterized a specific anthracnose resistance gene2 (*ARG2*), which plays a critical role in orchestrating defense responses against early anthracnose infection [266].

## 6. Advances in Research and Fungal Phytopathogen Management

### 6.1. Advances in Sorghum Breeding for Disease Resistance: MAS, CRISPR, and Transformation

These findings are being integrated into breeding programs to develop sorghum varieties with enhanced resistance. The use of marker-assisted selection (MAS) is particularly promising as it allows for the precise introgression of these resistance genes into susceptible sorghum varieties. This approach is supported by the identification of significant marker–trait associations that facilitate the selection of resistant genotypes [273]. Research efforts continue to explore the potential of CRISPR/Cas9 for developing resistant sorghum strains. These projects aim to target specific susceptibility genes and enhance the expression of resistance-related genes, leveraging the technology’s ability to introduce targeted mutations efficiently [274]. Recent advancements in plant transformation techniques have significantly improved the ability to introduce desirable traits into monocot crops, including sorghum. A notable example is the use of maize genes Baby Boom (*Bbm*) and Wuschel2 (*Wus2*), which have been shown to enhance transformation efficiencies in previously non-transferable maize inbred lines. This approach has been successfully extended to other monocots like sorghum, where the expression of these genes in immature embryos has facilitated the recovery of transgenic plants [275].

### 6.2. Predictive Modeling and Digital Agriculture: Role in Early Warning and Disease Management

Digital agriculture tools have emerged as essential in managing fungal diseases in sorghum, employing technologies such as predictive modeling, remote sensing, and IoT devices for early warning, detection, and management [276,277]. Predictive modeling and digital agriculture have significantly reshaped crop disease management, particularly in sorghum farming. Understanding the transformative impact of these technologies will enable early warning systems and the proactive management of fungal diseases, thereby considerably diminishing their adverse effects on crops [278,279]. These advancements facilitate early detection and intervention, which are crucial for preventing the escalation of diseases and reducing potential crop losses [280]. It further elaborated on the intricacies of predictive modeling, a technique that employs statistical methods and computational algorithms to forecast disease outbreaks [281]. Predictive models offer precise predictions of the likelihood of disease occurrence by analyzing a range of factors, including weather conditions, crop growth stages, and historical patterns of disease incidence [282]. This integration of predictive modeling into digital agriculture is equipped with sophisticated tools for efficient and effective disease management, paving the way for healthier crops and enhancing agricultural productivity [283]. This shift toward data-driven and responsive farming practices marks a significant advancement in agricultural technology and disease management strategies.

Early warning systems and optimized interventions are transforming the landscape of agricultural disease management. Emphasizing the importance of predictive models in offering early warnings regarding potential disease risks enables farmers to implement preventative measures before diseases become widespread [284]. This proactive approach is crucial for maintaining crop health and reducing disease outbreak severity [285]. These models can facilitate targeted and timely interventions by accurately forecasting the likelihood of disease occurrence in specific areas and at specific times [286]. This ensures more effective disease control and also significantly reduces the costs and environmental impacts associated with the excessive or misdirected use of treatments, such as fungicides [287]. In summary, the integration of warning systems and optimized interventions through predictive models is a significant step forward in making agricultural disease management more efficient, cost-effective, and environmentally sustainable. However, challenges such as data accuracy, access to technology, and farmer literacy remain challenging. Continued research and development, along with supportive policies, are crucial for overcoming these challenges and realizing the full potential of digital agriculture in managing fungal diseases in sorghum [288,289].

### 6.3. Fungal Phytopathogen Management Through Digital Agriculture

Digital agriculture encompasses various technologies that collect, process, and analyze data to guide decision making in agriculture. Remote sensing, precision agriculture, and farm management systems represent technological revolutions in agriculture, particularly disease management [290]. They emphasize the role of remote sensing technologies, such as drones, satellite imaging, and sensors, which enable real-time monitoring of crop health and environmental conditions, and provide critical data for early disease detection and prediction [291]. Complementing this illustrates advancements in precision agriculture, especially the use of variable rate technology. Variable rate technology allows for the tailored application of fungicides based on specific disease severity in different field areas, thereby enhancing disease control while minimizing resource wastage [292]. In addition, we focused on the integration capabilities of farm management systems (FMS). These digital platforms consolidate data from diverse sources, offer decision support through alerts of high disease risk, and provide recommendations for effective management [293]. Collectively, these innovative technologies mark a significant leap in agricultural practices, enhancing the efficiency, sustainability, and responsiveness of disease management in farming [294].

Data-driven decisions and reduced inputs are key components in revolutionizing agricultural disease management through digital agriculture (DA). This highlights the significant role of digital agriculture in facilitating data-driven decision making [295]. By offering accurate, timely, and localized data, digital technologies enable farmers to make informed decisions, thus enhancing the effectiveness and efficiency of disease management strategies [296]. This approach allows for the precise targeting of diseases, optimization of treatment plans, and improvement of overall crop health [297]. Concurrently, the benefits of precision application of fungicides and other agricultural inputs are enabled by digital technologies. This precision application significantly reduces the total quantity of inputs required, thereby lowering both the costs for farmers and the environmental impact [298]. The reduced use of inputs contributes to economic savings and aligns with sustainable agricultural practices. Collectively, these advancements in digital agriculture represent a major step toward creating sustainable, efficient, and effective disease management strategies [189]. However, promising predictive modeling and digital agriculture also face challenges. These include the accuracy of model access to data, technological digital literacy among farmers, and issues related to data privacy and ownership [299]. Continued research and development, capacity building, and supportive policies are needed to overcome these challenges and realize the full potential of these technologies for managing fungal diseases in sorghum and other crops.

## 7. Future Directions and Opportunities

### 7.1. Potential Improvements in Fungal Phytopathogen Management Strategies

The future of managing fungal diseases in sorghum appears promising with the convergence of various scientific and technological advancements [156]. Although these improvements hold great promise, their successful implementation requires continued investment in research and development, capacity building among farmers and extension workers, supportive policies, and institutional environments. Potential improvements in fungal phytopathogen management strategies are listed in Table 7.

### 7.2. Role of Technology and Digital Transformation in Improving Disease Detection and Management

The roles of technology and digital transformation in improving disease detection and management are continuously evolving. They offer significant potential for the early detection of diseases, more precise interventions, and improved decision-making [308,309]. However, challenges must be overcome to fully leverage the potential of technology and digital transformation. These include the digital divide, data privacy, security concerns, and the need for digital literacy among farmers. Integrating these technologies into coherent, user-friendly, and affordable solutions is an ongoing area of innovation and development. Key areas in which technology makes a difference are listed in Table 8.

### 7.3. Policy Recommendations for Supporting Farmers and Research in Disease Management

Policies play a crucial role in successfully managing fungal diseases in sorghum [318]. This can create an enabling environment for research, innovation, and the adoption of effective disease management practices. The following are some key policy recommendations [319]. Support from research and development is crucial for combating fungal diseases that affect agriculture. This emphasizes the need to increase public research and development investment [320]. This investment should cover a broad spectrum, including basic research to deepen the understanding of these diseases, applied research to tackle practical challenges, and the development of innovative technologies and solutions [321]. Furthermore, fostering collaboration and knowledge sharing among diverse stakeholders, such as researchers, farmers, and extension workers, is essential [322]. Such collaboration ensures that the research addresses the real needs of farmers, and that the results are effectively communicated and implemented in the field. This holistic approach can lead to more effective and sustainable management strategies for fungal diseases, thereby benefiting global agriculture [29,185,323].

Extension services and farm training are pivotal in enhancing agricultural productivity and sustainability. This highlights the importance of strengthening extension services [324]. These services are crucial for delivering timely, accurate, and localised information to farmers and guiding them toward effective disease management strategies. This involves disseminating knowledge and providing practical advice tailored to specific local conditions [325]. In addition, they advocate for significant investments in farmer training programs. Such programs are essential for equipping farmers with advanced knowledge and skills. The key focus areas include disease recognition, integrated pest management, precision agriculture, and other relevant practices [21,294]. Through these programs, farmers can learn how to apply innovative and efficient methods to their farming practices, leading to improved crop health and yield. Robust extension services and comprehensive farmer training can substantially contribute to the resilience and advancement of the agricultural sector [326].

Access to inputs and technologies is essential for empowering farmers, especially in managing and mitigating the effects of diseases on crops. Establishing effective seed systems is vital for ensuring the availability and accessibility of disease-resistant sorghum varieties [264]. These varieties can significantly enhance the resilience of sorghum crops to various diseases, thereby providing a staple food source in numerous regions [156]. Emphasis should be placed on the importance of facilitating farmers’ access to essential inputs and technologies. This includes easy access to fungicides and biocontrol agents, which are critical for disease management [70,327]. Moreover, introducing farmers to digital agricultural technologies can revolutionize their monitoring and management of crop health, leading to more precise and effective disease control [300].

We propose considering incentives or subsidies to encourage the adoption of sustainable and innovative disease management practices and technologies. This approach is essential for smallholder farmers who lack the resources to invest in advanced technologies [305,328]. By offering financial support or incentives, farmers can adopt more sustainable and effective disease management strategies to improve crop yield and overall livelihood. Overall, ensuring access to the right inputs and technologies is a key step toward sustainable agriculture and food security [329].

Regulatory policies in agriculture are vital for safeguarding both the environment and the efficacy of farming practices [330]. This emphasizes the need for science-based regulations to register and use inputs, such as fungicides, biocontrol agents, and genetically modified or genome-edited crops. These regulations ensure the safety and effectiveness of these technologies, thereby protecting both agricultural output and ecosystems [331]. This highlights the importance of data privacy and security policies in digital agricultural technologies. Such policies are essential for protecting the data of farmers, building trust, and facilitating the adoption of advanced technologies [332]. Collectively, these regulatory measures play a critical role in maintaining a sustainable, efficient, and safe agricultural sector that benefits farmers and the broader community [333].

Climate change and sustainability are critical factors that shape agricultural disease management strategies. We stress the importance of incorporating climate change considerations into these strategies, acknowledging that changing climatic conditions can significantly alter disease dynamics, thereby affecting the effectiveness of current management practices [334,335,336]. Advocates promote sustainable disease management practices, particularly for sorghum production. These practices should focus on reducing the reliance on chemical fungicides, which, while effective in the short term, can have detrimental environmental effects over time [248]. Emphasis should instead be placed on enhancing the resilience of crop production systems through sustainable methods such as biocontrol agent-integrated pest management and the development of disease-resistant varieties [45]. This approach ensures effective disease control as well as promotes environmental sustainability by addressing the challenges posed by climate change and securing the future of agriculture [337,338].

Economic and social policies are essential for reinforcing the sustainability and resilience of sorghum farming in the face of various challenges, including disease management. This highlights the importance of policies supporting the economic viability of sorghum farming [339]. These include implementing price-support mechanisms, providing crop insurance, and facilitating access to credit. Such policies can significantly enhance the ability of farmers to manage diseases effectively and mitigate potential financial losses from crop diseases [340]. Financial support is crucial for stabilizing the incomes of farmers and ensuring the continuity of farming operations.

In addition, it emphasizes the need for social policies that address broader social dimensions within the agricultural sector. These policies should focus on issues related to labor, gender, youth, and other social factors that influence the capacity and motivation of individuals engaged in sorghum farming [341]. By addressing these social dimensions, policies can create a more inclusive and supportive environment for all participants in the agricultural sector. This approach aids disease management as well as contributes to the overall development and sustainability of agricultural communities [342]. Collectively, these economic and social policies form a comprehensive support system that strengthens the resilience and productivity of sorghum farming, benefiting both farmers and the wider society. These policy recommendations aim to create an enabling environment for effective and sustainable sorghum disease management. However, it is important to note that policy development and implementation should be context-specific, considering local agricultural systems, socio-economic conditions, and institutional capacities. Moreover, stakeholder involvement and policy coherence across sectors and governance levels are critical to the success of these policies.

## 8. Conclusions

Fungal diseases in sorghum production pose significant challenges to agricultural productivity, food security, and livelihoods of farming communities globally. Advancements in technology, research, and innovative disease management strategies have provided substantial opportunities for improvement. An integrated disease management approach that combines traditional practices, such as crop rotation and intercropping, with modern techniques, such as using disease-resistant varieties, biological control, and precise fungicide applications, is pivotal. Implementing these strategies, backed by ongoing research and development in genomics, predictive modeling, and digital agriculture, can significantly bolster the fight against fungal diseases. Technology and digital transformation are instrumental in improving disease detection and management through remote sensing, IoT devices, predictive modeling and precision agriculture technologies, enabling more efficient, effective, and environmentally friendly disease management with early warning systems, precise interventions, and improved decision-making. Policy support is equally vital. Governments should support research and development, strengthen extension services, facilitate access to necessary inputs and technologies, and implement effective regulatory policies that respond to climate change challenges while supporting the economic and social well-being of farming communities. Although fungal diseases in sorghum remain a significant concern, the combined efforts of researchers, farmers, policymakers, and other stakeholders could effectively manage these diseases. By leveraging scientific and technological advancements, improving disease management practices, and providing enabling policies and institutional support, the resilience of sorghum production systems can be enhanced, thereby contributing to food security and sustainable rural development.

## Figures and Tables

**Figure 1 jof-11-00207-f001:**
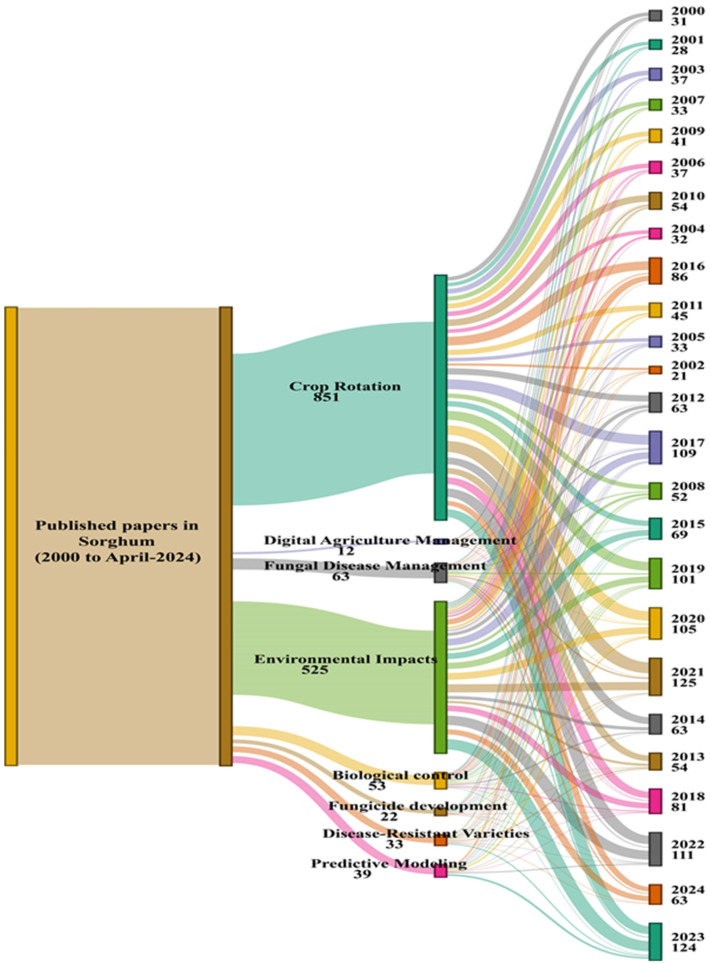
Timeline of research and review articles on sorghum that were published from 2000 to April 2024. Mining of this analysis to describe the total number of publications was published within the literature domain. The Web of Science database was searched using related keywords, and we found that 851 reports on crop rotation, 525 on environmental impacts, 63 on fungal disease management, 53 on biological control, 39 on predictive modeling, 33 on disease-resistant varieties, 22 on fungicide development, and 12 on digital agriculture management were published.

**Figure 2 jof-11-00207-f002:**
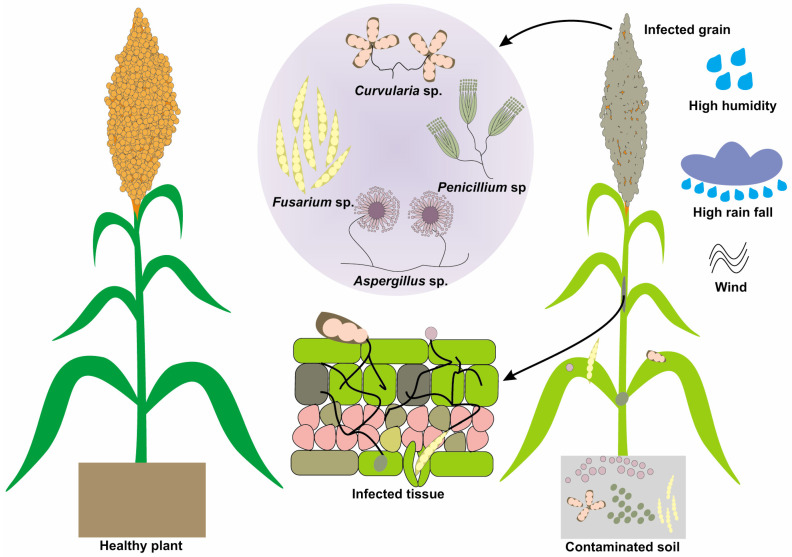
The symptoms of grain mold complex disease of sorghum and the intricate network of fungal hyphae enveloping grain particles. This complex symbiosis illustrates the interplay between fungi and grains in agricultural ecosystems.

**Figure 3 jof-11-00207-f003:**
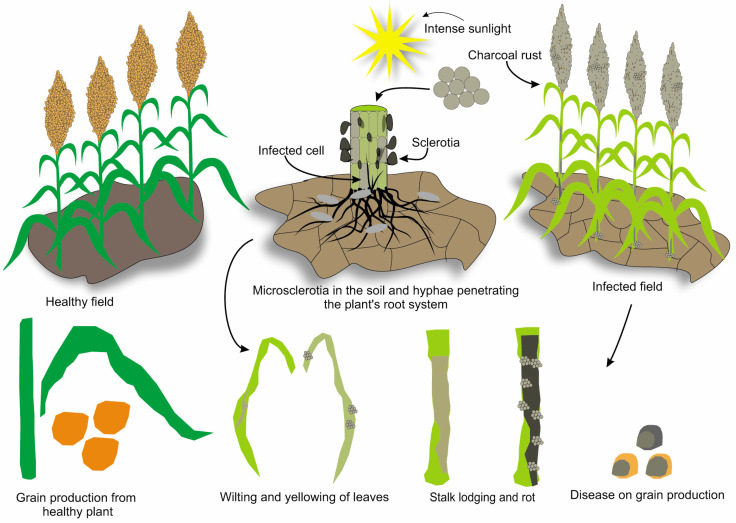
Charcoal rot disease symptoms on sorghum plants showcase characteristic discoloration and fungal growth in the stem tissues.

**Figure 4 jof-11-00207-f004:**
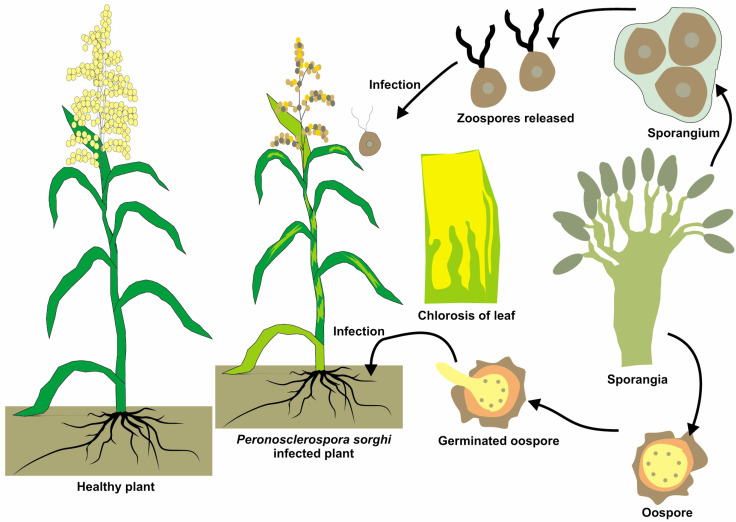
Downy mildew symptoms on sorghum leaves, characterized by yellowing and fuzzy grayish patches, caused by the fungal pathogen *Peronosclerospora sorghi*.

**Figure 5 jof-11-00207-f005:**
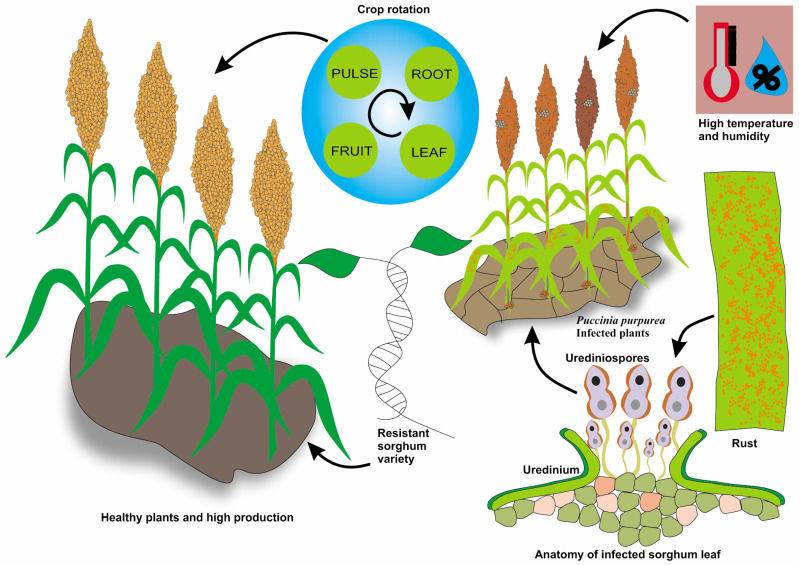
Rust disease on sorghum leaves. Orange pustules indicative of fungal infection are visible, accompanied by yellowing and necrosis of leaf tissue.

**Table 1 jof-11-00207-t001:** Management practices for controlling fungal phytopathogens in sorghum.

Practice	Key Points	Implications	References
Planting and Crop Rotation	Tilling or no-till methods, soil health, disease risk, crop rotation benefits	Balancing soil health and disease control, crop diversity	[53,54,55]
Irrigation and Drainage	Stable yield vs. fungal pathogen risk, wet condition diseases	Irrigation management, disease prevention in wet conditions	[29,56]
Fertilization	Nutrient management, nitrogen levels, plant health vs. fungal growth	Optimizing fertilization, balancing growth and disease resistance	[57,58,59]
Pest Management	Chemical, biological, and mechanical methods; insect–fungus interaction	Effective pest control to reduce disease spread	[60,61,62,63]
Harvest and Post-Harvest Practices	Harvest timing, pot-harvest handling, grain mold risk, disease inoculum	Reduced disease prevalence, effective crop residue management	[64,65,66]

**Table 2 jof-11-00207-t002:** Importance of managing fungal diseases in sorghum.

Aspect	Key Points	Implications	References
Yield Protection	Yield losses, Total crop failure	Farmer income protection, crop productivity	[18,67]
Grain Quality Maintenance	Reduced grain market value, grain mold effects	Market competitiveness, grain quality	[68,69]
Food Security	Staple food, impact on local food security	Community nutrition, dependence on sorghum	[70,71]
Feed and Industrial Uses	Usage in animal feed and biofuel, quality and availability impact	Industrial and feed sector reliance	[13,14,72]
Environmental Stewardship	Sustainable practices, reduced chemical use	Environmental health, biodiversity promotion	[20,21,73]
Economic Stability	Market stability, economic planning impact	Regional/national economic health	[74,75]

**Table 3 jof-11-00207-t003:** Other significant diseases in sorghum and their management strategies.

Fungal Phytopathogens	Key Symptoms	Impact	Prevalence (Percentage Contribution)	Key Management Strategies	References
Ergot (*Claviceps africana*)	Honeydew structures, sclerotia formation	Reduced yield and grain quality, toxic sclerotia	25%	Planting tolerant varieties, crop rotations with legumes	[83,84]
Head smut (*Sporisorium reilianum*)	Grain head replaced with spore mass	Significant yield loss, plants fail to produce grain	15%	Crop rotation, resistant varieties	[91,92,157]
Leaf blight (*Exserohilum turcicum*)	Rectangular, tan leaf lesions	Reduced yield from impaired photosynthesis, leaf death	20%	Agronomic cultural practices, the use of resistant or tolerant cultivars, biological control and chemical methods	[119,158]
Stalk rot (*Fusarium, macrophomina*)	Soft, rotten stalk base, plant lodging	Significant yield loss, especially when plants lodge	10%	Resistant varieties, agronomic practices	[27,120]
Sooty stripe (*Ramulispora sorghi*)	Yellow to tan leaf stripes, sooty spores	Variable yield impact, significant under high disease pressure	30%	Use protectant fungicides, cultural methods, seed soaking method, rotation of crops, manage proper planting distance	[139,140]

**Table 4 jof-11-00207-t004:** Additional common fungal phytopathogens in sorghum.

Fungal Phytopathogens	Key Symptoms	Impact	Stages of Plant Growth Most Affected	Management Strategies	References
Anthracnose (*Colletotrichum sublineolum*)	Water-soaked lesions on leaves, leaf/stem/panicle rot	Yield loss, infected plant debris as fungal source	Vegetative stage affects leaf and stem development, reproductive stage impacts panicle formation and grain development	Crop rotation, cultural practices, use resistant hybrid cultivars, use deep plowing leftover the crop residues from soil, biological control.	[93,159]
Grain Mold Complex (*Fusarium* spp., *Curvularia* spp., and others)	Grain discoloration, shriveling, quality reduction	Reduced grain quality, problematic in humid conditions	Grain filling stage affects kernel development, maturity stage increases susceptibility to mold formation	Maintain optimal plant population, crop adopting pest management practices, crop rotation, planting sorghum hybrids varieties, harvest panicles timely dry them quickly under natural sunlight, sort out moldy and damaged panicles, prevent insect damage stored grain suitable fumigation, monitor sorghum grain production, process and storage stages for mycotoxin contamination.	[31,33,160]
Charcoal rot (*Macrophomina phaseolina*)	Wilting, yellowing leaves, silver-gray stalk, microsclerotia	Significant yield loss, stalk lodging in drought conditions	Mid to late vegetative stages causes wilting and leaf yellowing, reproductive stage leads to stalk lodging, especially under drought stress	Use fungicides to inhibit mycelial growth, combination of soil solarization and organic amendment, controlling pathogens by soil mulching and large coverings with transparent polyethylene tarp, crop rotation, tillage practices, and reduce soil moisture.	[49,161]
Downy Mildew (*Peronosclerospora sorghi*)	White to purple downy growth on leaves, chlorosis, stunting	Yield loss, sterile heads, or no heads in severe cases	Vegetative stage causes leaf chlorosis and stunting, early reproductive stage reduces head formation and fertility	Use chemical, genetic, and cultural methods, use resistant sorghum varieties, seed treatments with the systemic fungicides metalaxyl and mefenoxam, cultural controls, crop rotation, deep tillage.	[113,121]
Rust (*Puccinia purpurea*)	Small, round pustules on leaves, leaf blight, early senescence	Reduced photosynthesis, impact on yield and grain quality	Vegetative stage affects leaf health and photosynthesis, reproductive stage impacts flowering and grain development	Cultivating resistant varieties, cultivating the slow-rusting sorghum varieties, use cultural practices, destroy infected residues from crop and weed hosts, use healthy seed for planting, treat seeds to prevent urediniospore seed-borne infection.	[97,138,162]

**Table 5 jof-11-00207-t005:** Innovative fungicide strategies.

Strategy	Key Features	Potential Benefits	References
Bio-based Fungicides	Natural substances, less environmental impact	Eco-friendly, safe for non-target organisms	[184,185]
Nano-formulations	Improved absorption, reduced doses, prolonged activity	Increased efficacy, targeted delivery, environmental safety	[186]
Fungicide Mixtures	Multiple fungicides, enhanced efficacy, resistance management	Improved disease control, reduced resistance risk	[187,188]
New Modes of Action	Novel action modes, controls resistant fungi	Effective against resistant strains, delays resistance development	[185,189]
Seed Treatment Fungicides	Protection from seed borne/pathogens, reduced foliar applications	Early protection, reduced chemical use	[36,190]

**Table 6 jof-11-00207-t006:** Genetic research strategies for disease resistance.

Strategy	Key Features	Potential Benefits	References
Quantitative Trait Loci Mapping	Mapping genes/gene regions, disease resistance	Efficient trait selection, complex trait management	[191,267]
Genome-Wide Association Studies	Genetic variant identification, resistance gene location	Understanding gene function and interaction	[157,268]
Transcriptomics	Gene expression analysis during infection	Identification of key resistance genes	[269]
Proteomics and Metabolomics	Defense response proteins and metabolites analysis	Mechanism revelation, new target identification	[184,270]
Genomic Selection	Performance prediction, genomic data utilization	Accelerated breeding process	[254,271]
Genome Editing	Precise gene editing, disease resistance enhancement	Direct modification of defense genes	[264,272]

**Table 7 jof-11-00207-t007:** Strategies for sorghum disease management.

Strategy	Key Features	Potential Benefits	Application	References
Integrate Disease Management	Genetic resistance, crop management, biological and chemical control	Improved effectiveness, sustainability	Fungal disease management in sorghum	[300,301]
Precision Farming	Real-time data, site-specific management, precision in fungicide application	Increased efficacy, cost reduction, Environmental protection	Site-specific sorghum farming practices	[279,302]
Breeding Technologies	Genomic selection, genome editing, advanced breeding	Efficient disease-resistant variety development	Breeding of sorghum varieties	[156,258]
Biological Control	Beneficial microbes, bio-fungicides, microbiome engineering	Enhanced natural plant defenses	Control of fungal diseases in sorghum	[185,303]
Predictive Models	Accuracy, accessibility, early warning	Optimized disease control, preventative measures	Disease management decision-making	[286,304]
Climate-Smart Agriculture	Resistant varieties, adapted crop management, climate adaptation	Resilience to climate change, disease resistance	Adapting sorghum farming to climate change	[300,305]
Farmer Training and Extension Services	Disease recognition education, localized extension services	Effective implementation, knowledge sharing	Farmer education and support	[306]
Policy and Institutional Support	Research support, incentives for sustainable practices, seed system establishment	Promotion of improved practices, policy support	Supporting sorghum disease management	[122,307]

**Table 8 jof-11-00207-t008:** Technologies in agricultural disease management and their applications.

Technology	Key Features	Potential Impact	Use Case	References
Remote Sensing and Imagery Analysis	Satellite imagery, drone-based sensing, machine learning analysis	Early disease detection	Disease identification	[290,310]
IoT and Sensor Technology	Temperature, humidity, rainfall, soil moisture monitoring	Risk prediction, informed decisions	Field condition monitoring	[101,311]
Predictive Modeling	Historical data analysis, predictive outbreak modeling	Early warning, preventative actions	Disease outbreak prediction	[49,312]
Precision Agriculture	Site-specific disease management, variable rate technology	Optimized fungicide use	Disease management	[294,313]
Mobile and Cloud-Based Apps	Real-time data, predictive models, decision support tools	Accessible information, enhanced communication	Farmer decision support	[61,314]
Artificial Intelligence and Machine Learning	Pattern detection, predictive insights, improved decision accuracy	Enhanced detection and prediction	Data analysis and insights	[278,315]
Block chain Technology	Secure data records, traceability, data integrity	Data security, traceability	Data management and security	[316]
Data Analytics and Decision Support Systems	Insight extraction, expert knowledge integration	Evidence-based recommendations	Decision support and analytics	[309,317]

## Data Availability

Not applicable.

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
