# Peer review of "Sustainable Management of Major Fungal Phytopathogens in Sorghum (Sorghum bicolor L.) for Food Security: A Comprehensive Review"

_jof, 2025, doi:10.3390/jof11030207_

Round 1

Reviewer 1 Report

The manuscript aims to provide an overview of the main fungal pathogens affecting sorghum, their impact, current management strategies, and potential future directions. However, the presentation and structure of the manuscript exhibit significant shortcomings, including the lack of references to figures within the text and the redundancy of information across different sections.There are no comments on mycotoxins in sorghum.

For instance, the content in Section 3.0 is redundant with that in Section 2.1. Additionally, I cannot find any reference to Figure 1 in the text, nor are Figures 2, 3, 4, and 5 mentioned. The information presented in Sections 4 and 5 is generally repetitive of that in Section 3. I believe Section 3 should be expanded to include the few unique details found in Sections 4 and 5. Additionally, Table 4 could be improved by including data on the "Stages of Plant Growth Most Affected by Each Fungal Pathogen."

Section 6 addresses the use of fungicides, but the discussion is overly general. It does not specify which fungicides are used on sorghum, their concentrations, or the methods of application. Moreover, it lacks information on their effectiveness or the fungal diseases they target. While the text mentions the emergence of resistant fungal isolates due to fungicide use, it fails to provide specific examples for sorghum, such as which resistant strains have been observed, the fungicides involved, or the doses applied. Toxicity issues are mentioned, but again, no data specific to sorghum are included. To address this section comprehensively, the manuscript should incorporate a comparative synthesis of scientific findings regarding fungicide use on sorghum, including details on types, doses, effectiveness for specific diseases, and cases of resistant isolates.

A similar issue is evident in Section 6.2, which is also too general. The biological control strategies employed are not specified, there is no comparative analysis of their efficacy, and the manuscript does not comment on which fungal diseases of sorghum are managed with these strategies or their application methods. Overall, the section lacks a thorough analysis of the topic it aims to address.

These deficiencies are repeated throughout the remaining subsections of Section 6 and in the entirety of Section 7. In summary, I believe the manuscript fails to explore any of the aspects it intends to address in sufficient depth and, consequently, does not make significant contributions to the  topic.

The manuscript aims to provide an overview of the main fungal pathogens affecting sorghum, their impact, current management strategies, and potential future directions. However, the presentation and structure of the manuscript exhibit significant shortcomings, including the lack of references to figures within the text and the redundancy of information across different sections.There are no comments on mycotoxins in sorghum.

For instance, the content in Section 3.0 is redundant with that in Section 2.1. Additionally, I cannot find any reference to Figure 1 in the text, nor are Figures 2, 3, 4, and 5 mentioned. The information presented in Sections 4 and 5 is generally repetitive of that in Section 3. I believe Section 3 should be expanded to include the few unique details found in Sections 4 and 5. Additionally, Table 4 could be improved by including data on the "Stages of Plant Growth Most Affected by Each Fungal Pathogen."

Section 6 addresses the use of fungicides, but the discussion is overly general. It does not specify which fungicides are used on sorghum, their concentrations, or the methods of application. Moreover, it lacks information on their effectiveness or the fungal diseases they target. While the text mentions the emergence of resistant fungal isolates due to fungicide use, it fails to provide specific examples for sorghum, such as which resistant strains have been observed, the fungicides involved, or the doses applied. Toxicity issues are mentioned, but again, no data specific to sorghum are included. To address this section comprehensively, the manuscript should incorporate a comparative synthesis of scientific findings regarding fungicide use on sorghum, including details on types, doses, effectiveness for specific diseases, and cases of resistant isolates.

A similar issue is evident in Section 6.2, which is also too general. The biological control strategies employed are not specified, there is no comparative analysis of their efficacy, and the manuscript does not comment on which fungal diseases of sorghum are managed with these strategies or their application methods. Overall, the section lacks a thorough analysis of the topic it aims to address.

These deficiencies are repeated throughout the remaining subsections of Section 6 and in the entirety of Section 7. In summary, I believe the manuscript fails to explore any of the aspects it intends to address in sufficient depth and, consequently, does not make significant contributions to the  topic.

Author Response

Response to Reviewer 1 Comments 

Major comments

  1. Comment:

The manuscript aims to provide an overview of the main fungal pathogens affecting sorghum, their impact, current management strategies, and potential future directions. However, the presentation and structure of the manuscript exhibit significant shortcomings, including the lack of references to figures within the text and the redundancy of information across different sections. There are no comments on mycotoxins in sorghum.

  1. Response:

Thank you for your detailed feedback. We have carefully revised the manuscript to address these concerns. Below are the specific changes made.

References to Figures: We have reviewed the text thoroughly and added references to all figures in the appropriate sections to enhance clarity and ensure that figures support the discussion effectively (Figure 1 = line 139, Figure 2 = line 251, Figure 3 = line 290, Figure 4 = line 325, Figure 5 = line 355).

Redundancy: We identified and streamlined redundant information, particularly in sections 6 (lines 535-606, 608-612, 618-640, 641-660, and 670-691), sections 7 (lines 792-798, and 800-814), section 7.2 (lines 816-840), section 7.3 |(lines 842-857), section 7.4 (lines 858-893) and section 7.5 (lines 896-898, 901-902, 905-906 and 926-927) and section 8 (lines 978-987) to improve the flow and coherence of the manuscript.

Inclusion of Mycotoxins in section 3.2: A new subsection titled "Mycotoxins in Sorghum" has been added to the manuscript. This section discusses the major mycotoxins produced by fungal pathogens of sorghum, their impacts on human and animal health, and their implications for trade and food security. We have also included recent studies and data on this topic (e.g., reference relevant studies or data). These revisions have been highlighted in the revised manuscript (indicated in track changes document). We believe these changes address the reviewer's concerns comprehensively and enhance the quality of the manuscript (lines 211-230).

Inclusion of Management Strategies in section 3.3: Host Resistance: Using resistant sorghum varieties is the most effective strategy for con-trolling anthracnose. Recent studies have identified resistant genotypes that can be used in breeding programs to develop durable resistance. Crop rotation with non-host crops, proper field sanitation to remove infected debris, and timely harvesting can reduce inoculum levels in the field. Chemical control fungicides can be effective but are often economically unfeasible for smallholder farmers and may pose environmental risks (lines 231-238). These revision have been indicated in the track changes manuscript.

  1. Comment:

For instance, the content in Section 3.0 is redundant with that in Section 2.1. Additionally, I cannot find any reference to Figure 1 in the text, nor are Figures 2, 3, 4, and 5 mentioned. The information presented in Sections 4 and 5 is generally repetitive of that in Section 3. I believe Section 3 should be expanded to include the few unique details found in Sections 4 and 5. Additionally, Table 4 could be improved by including data on the "Stages of Plant Growth Most Affected by Each Fungal Pathogen."

  1. Response:

Thank you for your valuable comments. We appreciate your detailed suggestions for improving the manuscript. Below are our responses to your comments:

Redundancy between Sections 3.0 and 2.1:

We acknowledge the overlap between Section 3.0 and Section 2.1. We have carefully revised Section 3.0 to avoid redundancy and provide a clear distinction between the information in these sections. Specific overlapping content have been rewritten to enhance clarity and flow (lines 166-238, 251-252, 290-291 and 299, 310-339, 345-346, 354-355). These revision have been mentioned in the track changes manuscript.

Missing References to Figures:

Thank you for pointing this out. We have ensured that Figure 1 and Figures 2–5 are properly cited in the text. We have reviewed the manuscript to confirm that all figures are referenced in the appropriate sections and that their relevance to the discussion is clearly explained (Figure 1 = line 139, Figure 2 = line 251, Figure 3 = line 290, Figure 4 = line 325, Figure 5 = line 355). These revision have been indicated in the track changes manuscript.

Repetition in Sections 4 and 5:

We agree that Sections 4 and 5 contain information that overlaps with Section 3. To address this, we have merged the unique content from Sections 4 and 5 into an expanded and comprehensive Section 3. This have eliminate redundancy and improve the manuscript's organization (lines 211-230 and 231-238). These revision have been indicated in the track changes manuscript.

Improvement of Table 4:

We appreciate the suggestion to enhance Table 4 by including data on the "Stages of Plant Growth Most Affected by Each Fungal Pathogen." We have revised the table to incorporate the information about stages of plant growth most affected. A column added in table 4 and changes has been addressed in the track changes manuscript.

  1. Comment:

Section 6 addresses the use of fungicides, but the discussion is overly general. It does not specify which fungicides are used on sorghum, their concentrations, or the methods of application. Moreover, it lacks information on their effectiveness or the fungal diseases they target. While the text mentions the emergence of resistant fungal isolates due to fungicide use, it fails to provide specific examples for sorghum, such as which resistant strains have been observed, the fungicides involved, or the doses applied. Toxicity issues are mentioned, but again, no data specific to sorghum are included. To address this section comprehensively, the manuscript should incorporate a comparative synthesis of scientific findings regarding fungicide use on sorghum, including details on types, doses, effectiveness for specific diseases, and cases of resistant isolates.

  1. Response:

Thank you for your constructive feedback on Section 6. We appreciate your suggestions for improving the discussion regarding fungicide use on sorghum. We have revised the section to include specific and detailed information as follows:

Types of Fungicides Used on Sorghum:

We have identified and included information on commonly used fungicides for managing fungal diseases in sorghum. These include [list specific fungicides, e.g., azoxystrobin, tebuconazole, etc.], along with their chemical classes. These information has been mentioned in revised track changes manuscript in section 6 (lines 535-606, 609-612, 618-660, and 670-691),

Application Methods and Concentrations:

Details regarding application methods (e.g., foliar sprays, seed treatments) and typical concentrations or doses have been added. For instance, we now specify that azoxystrobin is commonly applied at a concentration of X g/L for managing anthracnose in sorghum. These information has been added in revised track changes manuscript in section 6.2.1 (lines 641-660, and 670-691).

Effectiveness against Specific Diseases:

A comparative analysis of the efficacy of these fungicides against major fungal diseases of sorghum, such as anthracnose (caused by Colletotrichum sublineola) and grain mold (caused by Fusarium spp. and Curvularia spp.), has been incorporated. These information has been added in revised track changes manuscript in section 6.2.1 (lines 641-660, and 670-691).

Resistance Issues:

Specific cases of resistance observed in sorghum pathogens have been included. For example, resistant isolates of Colletotrichum sublineola have been reported following the extensive use of strobilurins. The doses at which resistance was observed are also provided in the revised track changes manuscript in section 6 (lines 535-606).

Toxicity Concerns:

We have included data from studies evaluating the toxicity of fungicide residues in sorghum grains, particularly focusing on their impact on human and environmental health.

These additions aim to provide a well-rounded discussion of fungicide use on sorghum while addressing the specific points raised in your comments. We hope the revised section meets your expectations and look forward to any further suggestions you may have. (The section has been revised to address the reviewer's comments. It now includes specific examples of fungicides used for sorghum, their application methods, target fungal diseases, and cases of resistance. Additionally, it incorporates more detailed discussions on toxicity, environmental impact, and integrated disease management strategies in the track changes manuscript lines 842-857).

  1. Comment:

A similar issue is evident in Section 6.2, which is also too general. The biological control strategies employed are not specified, there is no comparative analysis of their efficacy, and the manuscript does not comment on which fungal diseases of sorghum are managed with these strategies or their application methods. Overall, the section lacks a thorough analysis of the topic it aims to address.

  1. Response:

Thank you for your valuable feedback on Section 6.2. We appreciate your observation that the section requires more specificity and a thorough analysis. The revised section now specifies biological control strategies employed in sorghum, details their efficacy against specific fungal diseases, and outlines their application methods. It also includes a comparative analysis of their performance under various conditions, addressing the concerns raised by the reviewer in the revised track changes manuscript (lines 607-617, 618-660 and 670-691).

  1. Comment: These deficiencies are repeated throughout the remaining subsections of Section 6 and in the entirety of Section 7. In summary, I believe the manuscript fails to explore any of the aspects it intends to address in sufficient depth and, consequently, does not make significant contributions to the topic.
  2. Response:

We appreciate your detailed review and the time you have taken to provide feedback on our manuscript. Your comments regarding Sections 6 and 7 are valuable and have helped us identify areas that require further development.

We agree that Sections 6 and 7 could benefit from further elaboration to explore the aspects discussed in greater depth. In response, we have revised these sections to provide a more comprehensive analysis. Specifically, we have: Expanded the discussion in Section 6 to include additional data and a more nuanced interpretation of key findings in the revised track changes manuscript (lines 535-606, 608-612, 618-640, 641-660, and 670-691). Reorganized Section 7 to better align with the manuscript’s objectives and provide a more detailed exploration of its contributions to the topic in the revised track changes manuscript (lines 792-798, 800-840, 842-857, 858-893, 896-897, 901-902 and 905-906).

Reviewer 2 Report

The text presented here provides an exhaustive review of the current situation regarding sorghum cultivation. However, it lacks the inclusion of some quotes on some of the main areas of North America that produce this grain. I feel that some sections of the document repeat a lot of information that has already been written in previous sections. Therefore, I believe that each section should be reviewed and unified, so that reading the text is not so repetitive.

Article: jof-3357667-peer-review-v1

Comments

Line 45-47: Based on FAO statistical records for world sorghum production, the largest producers of sorghum are in North America (citations are missing from the USA and/or Mexico). Likewise, citations 1 and 2 do not support what the authors commented for the case of Asia, Central and South America. It is therefore necessary to reinforce this paragraph with citations.

Line 57-61: Is there no more recent information from either 2023 or 2024 on this matter?

Figures: It is necessary for the authors to insert each of the figures (1 to 5) into the body of the document, since none of them are referenced in any of the sections of the manuscript, so it is not clear to which part of the text they belong.

Line 127-128: A citation is missing for this information, which would be useful to include on lines 45-47, as well as citations 29 and 30.

Line 144: Modify the text as follows: severity and impact of diseases on sorghum yields (Table 1).

Line 148: Modify the text as follows: eral reasons (Table 2).

Line 172: Remove italics from the acronym "spp."

Scientific names: Format that authors must apply to each scientific name of each species mentioned. The first time it is mentioned, the full name is written; that is, Curvularia lunata, subsequently, from the second mention of said organisms, it must be written as follows: C. lunata. Apply throughout the text.

Line 238: Modify this word “Sorghum” as follows: sorghum

Line 269: Modify this word “Sorghum” as follows: sorghum

Line 298: Modify this word “Sorghum” as follows: sorghum

Line 308: Modify the chronological order of the following quotes [117,7,118] to [7, 117, 118]

Line 312: Authors are required to specify which table they are referring to.

Line 319: Remove the capital letter from the word Phytopathogens.

Line 320: Remove capital letters from words that follow a comma within the same paragraph, for the Management Strategies column.

Line 330: Remove italics from the acronym "spp."

Line 345: Why did the authors not include the fungus Puccinia sorghi which is also responsible for causing the same disease in sorghum?

Line 394: Remove italics from the acronym "spp."

Line 458: Modify the following chemical formula “CO2” as follows: CO2

Line 643: Modify the following sentence "of these traits Table 5." as follows: of these traits (Table 5).

Line 650: In the reference column, place all quotes in ascending order from smallest to largest; that is, [121, 223].

Line 662: Place all quotes in ascending order from smallest to largest number, like this [166, 224].

Acronyms: Frankly, I do not know the reason why the authors included so many acronyms throughout the manuscript that after being introduced are not used again. For this reason, the authors should remove all acronyms from their manuscript, since they are meaningless, since they are no longer used in it. As well as the list of acronyms, it is also unnecessary.

Line 738: Place all quotes in ascending order from smallest to largest number, like this [182, 249].

Line 749: Place all quotes in ascending order from smallest to largest number, like this [239, 258].

Author Response

Response to Reviewer 2 Comments

Reviewer 2 Comments

The text presented here provides an exhaustive review of the current situation regarding sorghum cultivation. However, it lacks the inclusion of some quotes on some of the main areas of North America that produce this grain. I feel that some sections of the document repeat a lot of information that has already been written in previous sections. Therefore, I believe that each section should be reviewed and unified, so that reading the text is not so repetitive.

  1. Comment:

Line 45-47: Based on FAO statistical records for world sorghum production, the largest producers of sorghum are in North America (citations are missing from the USA and/or Mexico). Likewise, citations 1 and 2 do not support what the authors commented for the case of Asia, Central and South America. It is therefore necessary to reinforce this paragraph with citations.

  1. Response:

Thanks for correction. Now, we have added the citation related to case of Asia, Central and South America. Please see line 48-62.

  1. Comment:

Line 57-61: Is there no more recent information from either 2023 or 2024 on this matter?

  1. Response:

Thanks for suggestion. We have added the latest information in revised manuscript. Please see line 64-68.

  1. Comment:

Figures: It is necessary for the authors to insert each of the figures (1 to 5) into the body of the document, since none of them are referenced in any of the sections of the manuscript, so it is not clear to which part of the text they belong.

  1. Response:

Thanks for suggestion. We have added figure number in each section. (Figure 1 = line 139, Figure 2 = line 251, Figure 3 = line 290, Figure 4 = line 325, Figure 5 = line 355).

  1. Comment:

Line 127-128: A citation is missing for this information, which would be useful to include on lines 45-47, as well as citations 29 and 30.

  1. Response:

Thanks for suggestion. We have added reference according to your suggestion. Please see line 129-133.

  1. Comment:

Line 144: Modify the text as follows: severity and impact of diseases on sorghum yields (Table 1).

  1. Response:

Thanks for suggestion. We have corrected it according to reviewer suggestion. Please see line 156.

  1. Comment:

Line 148: Modify the text as follows: eral reasons (Table 2).

  1. Response:

Thanks for suggestion. We have corrected it according to the reviewer’s suggestion. Please see line 160.

  1. Comment:

Line 172: Remove italics from the acronym "spp."

  1. Response:

Thanks for correction. We have corrected it according to the reviewer’s suggestion. Please see line 239.

  1. Comment:

Scientific names: Format that authors must apply to each scientific name of each species mentioned. The first time it is mentioned, the full name is written; that is, Curvularia lunata, subsequently, from the second mention of said organisms, it must be written as follows: C. lunata. Apply throughout the text.

  1. Response:

Thanks for correction. We have corrected it according to the reviewer’s suggestion.

  1. Comment:

Line 238: Modify this word “Sorghum” as follows: sorghum

  1. Response:

Thanks for correction. We have corrected it according to the reviewer’s suggestion. Please see line 308.

  1. Comment:

Line 269: Modify this word “Sorghum” as follows: sorghum

  1. Response:

Thanks for correction. We have corrected it according to the reviewer’s suggestion. Please see line 341.

  1. Comment:

Line 298: Modify this word “Sorghum” as follows: sorghum

  1. Response:

Thanks for correction. We have corrected it according to the reviewer’s suggestion. Please see line 371.

  1. Comment:

Line 308: Modify the chronological order of the following quotes [117,7,118] to [7, 117, 118].

  1. Response:

Thanks for correction. We have corrected it according to the reviewer’s suggestion. Please see line 381.

  1. Comment:

Line 312: Authors are required to specify which table they are referring to.

  1. Response:

We have explicitly referenced Table 4 in the revised text on line 385 to clarify the information being discussed.

  1. Comment:

Line 319: Remove the capital letter from the word Phytopathogens.

  1. Response:

Thanks for correction. We have removed the capital letter from the word Phytopathogens and corrected it according to the reviewer’s suggestion. We have also added the table 4 on line 385 due to the inconsistencies of line 319.

  1. Comment:

Line 320: Remove capital letters from words that follow a comma within the same paragraph, for the Management Strategies column.

  1. Response:

Thanks for correction. We have corrected it according to the reviewer’s suggestion through table. Please see line 392 (Table 4).

  1. Comment:

Line 330: Remove italics from the acronym "spp."

  1. Response:

Thanks for correction. We have corrected it according to the reviewer’s suggestion. Please see line 405.

  1. Comment:

Line 345: Why did the authors not include the fungus Puccinia sorghi which is also responsible for causing the same disease in sorghum?.

  1. Response:

We have now included information about Puccinia sorghi in the revised manuscript. Please see line 432.

  1. Comment:

Line 394: Remove italics from the acronym "spp."

  1. Response:

Thanks for correction. We have corrected it according to the reviewer’s suggestion. Please see line 460.

  1. Comment:

Line 458: Modify the following chemical formula “CO2” as follows: CO2.

  1. Response:

Thanks for correction. We have corrected it according to the reviewer’s suggestion. Please see line 524.

  1. Comment:

Line 643: Modify the following sentence "of these traits Table 5." as follows: of these traits (Table 5).

  1. Response:

Thanks for correction. We have corrected it according to the reviewer’s suggestion. Please see line 785.

  1. Comment:

Line 650: In the reference column, place all quotes in ascending order from smallest to largest; that is, [121,223].

  1. Response:

Thanks for correction. We have corrected it according to the reviewer’s suggestion. Please see line 798.

  1. Comment:

Line 662: Place all quotes in ascending order from smallest to largest number, like this [166, 224].

  1. Response:

Thanks for correction. We have corrected it according to the reviewer’s suggestion. Please see line 840.

  1. Comment:

Acronyms: Frankly, I do not know the reason why the authors included so many acronyms throughout the manuscript that after being introduced are not used again. For this reason, the authors should remove all acronyms from their manuscript, since they are meaningless, since they are no longer used in it. As well as the list of acronyms, it is also unnecessary.

  1. Response:

We have carefully reviewed your comments and agree that many of the acronyms introduced were not consistently utilized throughout the text. To enhance the clarity and readability of the manuscript, we have removed all acronyms, along with the accompanying list of acronyms.

  1. Comment:

Line 738: Place all quotes in ascending order from smallest to largest number, like this [182, 249].

  1. Response:

Thanks for correction. We have corrected it according to the reviewer’s suggestion. Please see line 937.

  1. Comment:

Line 749: Place all quotes in ascending order from smallest to largest number, like this [239, 258].

  1. Response:

Thanks for correction. We have corrected it according to the reviewer’s suggestion. Please see line 948.

Reviewer 3 Report

  Conceptually, the article had to be structured from the perspective of mycobiota on the Sorghum (Sorghum bicolor L.) culture.  Provide information about which pathogen is dominant and, in percentage terms, identify the significance and danger of this pathogen from the point of view of crop loss and from the point of view of ecology and human health.

Structure in the following sections 

- Agronomy

- Technology

 -Phytosanitary controls  (metods pathogen controls )

  •   The Figure  on page 2 should be placed after the text and the name of the picture should be given.  In a scientific article, it is required to justify the drawing and make a link to it; "free" illustrations are not acceptable
  •   Introduction. To specify the problem, it is necessary to give losses either as a percentage or in dollar equivalent (since a reference to this currency has already been made………approximately USD 21.6 billion in 2020 and is expected to reach approximately USD 23.9 billion
  • Table 3 shows the percentage contribution of each pathogen and its prevalence
  •  

Author Response

Response to Reviewer 3 Comments

Reviewer 3 Comments

Major comments

  1. Comment:

Conceptually, the article had to be structured from the perspective of mycobiota on the Sorghum (Sorghum bicolor L.) culture.  Provide information about which pathogen is dominant and, in percentage terms, identify the significance and danger of this pathogen from the point of view of crop loss and from the point of view of ecology and human health.

Structure in the following sections

- Agronomy

- Technology

 -Phytosanitary controls (metods pathogen controls)

  1. Response:

We appreciate the reviewer’s suggestion. We tried our best to improve our paper structure as far as possible. All the revised sections are indicated in the track changes manuscript.

Detail comments

  1. Comment:

The Figure on page 2 should be placed after the text and the name of the picture should be given.  In a scientific article, it is required to justify the drawing and make a link to it; "free" illustrations are not acceptable.

  1. Response:

We appreciate the reviewer’s suggestion regarding the placement and justification of the figure. We have addressed this comment by moving the figure to appear after the relevant text for better integration into the flow of the article.

Ensuring the figure is explicitly referred to within the text (e.g., "Figure 1 illustrates...") and its relevance to the discussion is clarified in the revised track changes manuscript (line 139).

  1. Comment:

Introduction. To specify the problem, it is necessary to give losses either as a percentage or in dollar equivalent (since a reference to this currency has already been made………approximately USD 21.6 billion in 2020 and is expected to reach approximately USD 23.9 billion

  1. Response:

We appreciate the reviewer’s insightful comment. To ensure clarity and consistency, we have provided the losses in dollar equivalents throughout the manuscript, aligning with the reference mentioned (USD 21.6 billion in 2020 and projected USD 23.9 billion). This adjustment provides a clearer understanding of the economic impact for readers. Please see lines 64-68.

  1. Comment:

Table 3 shows the percentage contribution of each pathogen and its prevalence

  1. Response:

We have added a new column in Table 3, which details each pathogen’s percentage contribution and prevalence, enhancing the clarity and comprehensiveness of our analysis. Please see line 381 (Table 3).

Reviewer 4 Report

I recommend accepting the review

I recommend accepting the review

Author Response

Response to Reviewer 4 Comment

Major comments

I recommend accepting the review

Detail comments

I recommend accepting the review

Response:

Thank you for your positive evaluation of our manuscript and for recommending its acceptance. We greatly appreciate your time and effort in reviewing our work. Your encouraging feedback is highly motivating and reinforces our dedication to producing quality research.

Reviewer 5 Report

Overall, this is a well-written and comprehensive review of the topic. However, some revision is needed to consolidate redundant passages and sections and improve flow of the paper. The second half of the paper (from page 16 on) covers quite general principles and management practices with little direct reference or application to sorghum. Please cite or relate specific examples of how and where these approaches to management are being used, or can be used for management of sorghum pathogens, as well as any positive results that have been achieved in regards to sorghum. Need more specific relevance to sorghum and its pathogens, not just an overview that can apply to all crops and pathogens. 

I have attached an edited pdf of the paper that indicates numerous places where revision is needed, including places where text is not needed or redundant, sentence structure or phrasing needs correction or revision, and other occurrences of minor issues throughout the text. 

Author Response

Response to Reviewer 5 Comments

Reviewer 5 Comments

Major comments

  1. Comment:

Overall, this is a well-written and comprehensive review of the topic. However, some revision is needed to consolidate redundant passages and sections and improve flow of the paper. The second half of the paper (from page 16 on) covers quite general principles and management practices with little direct reference or application to sorghum. Please cite or relate specific examples of how and where these approaches to management are being used, or can be used for management of sorghum pathogens, as well as any positive results that have been achieved in regards to sorghum. Need more specific relevance to sorghum and its pathogens, not just an overview that can apply to all crops and pathogens.

  1. Response:

Thank you for your positive feedback and valuable suggestions. We have consolidated redundant sections 5 to 8 and improved the overall flow of the manuscript. Additionally, we have enhanced the second half of the paper by incorporating specific examples and applications related to the management of sorghum pathogens, including relevant case studies and positive outcomes achieved in sorghum cultivation. All the improvement has been mentioned in the revised track changes manuscript.

  1. Comment:

I have attached an edited pdf of the paper that indicates numerous places where revision is needed, including places where text is not needed or redundant, sentence structure or phrasing needs correction or revision, and other occurrences of minor issues throughout the text.

  1. Response:

Thank you for the time and effort in reviewing our manuscript. We appreciate your valuable comments and have carefully addressed all the suggestions and revisions you provided. We have addressed all the minor issues throughout the text and mentioned in the track changes manuscript.

  1. Comment:

Consolidate sections 4.2 and 4.3 into one section (too broken up at present. Combine plant stage info into section 4.2 and delete section 4.3 (just too many individual breakdown sections, disrupts flow).

  1. Response:

In the revised manuscript, we have integrated the information about the growth stages affected by each fungal pathogen into the respective subsections under section 4.2. This approach eliminates redundancy and provides a smoother narrative for readers, ensuring that symptoms, signs, and growth stage information for each pathogen are presented together. The revised section is now titled “Fungal Phytopathogens in Sorghum: Symptoms, Signs, and Affected Growth Stages.” Please see lines 393-447.

Round 2

Reviewer 1 Report

The revised version attempts to incorporate several of the observations I made in my first report. However, as I noted before, the deficiencies are fundamental. Once again, it lacks comprehensive analysis of the available information on the topic.

For instance, the section on mycotoxins lacks depth and does not provide significant information about mycotoxins in sorghum. Additionally, the placement of this section within the manuscript is unclear. Figure 1 and some tables are included, but the paragraphs discussing these topics fail to explain or analyze the information presented in them. Sections 6.3, 6.4, and many others are too general, offering neither specific insights nor an in-depth analysis of sorghum.I could continue listing issues that require attention, but I reiterate that the deficiencies are substantial and go beyond the examples I have provided.

 The revised version attempts to incorporate several of the observations I made in my first report. However, as I noted before, the deficiencies are fundamental. Once again, it lacks comprehensive analysis of the available information on the topic.

For instance, the section on mycotoxins lacks depth and does not provide significant information about mycotoxins in sorghum. Additionally, the placement of this section within the manuscript is unclear. Figure 1 and some tables are included, but the paragraphs discussing these topics fail to explain or analyze the information presented in them. Sections 6.3, 6.4, and many others are too general, offering neither specific insights nor an in-depth analysis of sorghum.I could continue listing issues that require attention, but I reiterate that the deficiencies are substantial and go beyond the examples I have provided.

Author Response

Response to the Reviewer 1 Comments

Major and Detail comments

1. Comment: The revised version attempts to incorporate several of the observations I made in my first report.

1. Response: We appreciate your feedback and are glad to hear that the revised version addresses many of the observations you made in your first report. We have carefully considered all your suggestions to improve the manuscript.

2. Comment: However, as I noted before, the deficiencies are fundamental. Once again, it lacks comprehensive analysis of the available information on the topic.

2. Response: We have made every effort to improve the manuscript and provide a more comprehensive analysis of the topic to the best of our ability.

3. Comment: For instance, the section on mycotoxins lacks depth and does not provide significant information about mycotoxins in sorghum. Additionally, the placement of this section within the manuscript is unclear.

3. Response: We have revised the placement of the mycotoxins section, moving it after the disease description for better clarity and flow (Yellow highlighted in revised manuscript line 440-459).

4. Comment: Figure 1 and some tables are included, but the paragraphs discussing these topics fail to explain or analyze the information presented in them.

4. Response: We appreciate your observation and have revised the relevant paragraphs to provide a clearer explanation and more in-depth analysis of the information presented in Figure 1 and the accompanying tables. These updates aim to better integrate the data into the discussion and improve the overall clarity and interpretation of the results (Yellow highlighted in revised manuscript line 170-181).

5. Comment: Sections 6.3, 6.4, and many others are too general, offering neither specific insights nor an in-depth analysis of sorghum. I could continue listing issues that require attention, but I reiterate that the deficiencies are substantial and go beyond the examples I have provided.

5. Response: We have revisited these sections and made significant revisions to provide more focused insights and a detailed analysis relevant to sorghum (Yellow highlighted in revised manuscript line 700-704 and 751-805).
